# Shipborne Wind Measurement and Motion-induced Error Correction of a Coherent Doppler Lidar over the Yellow Sea in 2014

Xiaochun. Zhai[1], Songhua. Wu[1, 2], Bingyi. Liu[1, 2], Xiaoquan. Song[1, 2], Jiaping. Yin[3]

[1]Ocean Remote Sensing Institute, Ocean University of China, Qingdao, 266100, China

[2]Laboratory for Regional Oceanography and Numerical Modelling, Qingdao National Laboratory for Marine Science and Technology, Qingdao, 266100, China

[3]Seaglet Environmental Technology, Qingdao, 266100, China

*Correspondence to: S. Wu (wush@ouc.edu.cn)*

**Abstract.** Shipborne wind observations by a Coherent Doppler Lidar (CDL) have been conducted to study the structure of the Marine Atmospheric Boundary Layer (MABL) during the 2014 Yellow Sea campaign. This paper evaluates uncertainties associated with the ship motion and presents the correction methodology regarding lidar velocity measurement based on modified 4-Doppler Beam Swing (DBS) solution. The errors of calibrated measurement, both for the anchored and the cruising shipborne observations, are comparable to those of ground-based measurements. The comparison between the lidar and radiosonde results in a bias of -0.23 $ms^{-1}$ and a standard deviation of 0.87 $ms^{-1}$ for the wind speed measurement, and $2.48^{\circ}$, $8.84^{\circ}$ for the wind direction. The biases of horizontal wind speed and random errors of vertical velocity are also estimated using the error propagation theory and frequency spectrum analysis, respectively. The results show that the biases are mainly related to the measuring error of the ship velocity and lidar pointing error, and the random errors are mainly determined by the Signal-to-Noise Ratio (SNR) of lidar backscattering spectrum signal. It allows for the retrieval of vertical wind, based on one measurement, with random error below 0.15 $ms^{-1}$ for appropriate SNR threshold and bias below 0.02 $ms^{-1}$. The combination of the CDL attitude correction system and the accurate motion correction process has the potential of continuous long-term high temporal and spatial resolution measurement for MABL thermodynamic and turbulence process.

## 1 Introduction

The vertical structure of atmospheric variables in the Marine Atmospheric Boundary Layer (MABL) plays an important role in the earth's climate system, governing exchanges of energy, sensible heat, water vapour and momentum between ocean and the overlying atmosphere (Rocers et al., 1995; Wulfmeyer and Janjic, 2005), and the turbulence characteristics are significant for understanding the driving and coupling mechanisms and for parameterizing ocean-atmosphere interaction process. There are a number of studies on the turbulent fluxes measurement over the sea surface. Various motion sensing technique on the moving platform has been developed in the field of airborne (Axford, 1968), space-borne (Hawley et al., 1993) and shipborne observations (Fujitani, 1992; Song et al., 1996; Edson et al., 1998; Miller et al., 2008). Many shipborne field experiments have

been widely carried out over Pacific Oceanic area (Mitsuta et al., 1974; Bradley et al., 1991; Shao, 1995; Tsukamoto et al., 1995). One of the most common direct techniques for measuring surface fluxes is eddy-correlation, which utilizes the covariance of mixing ratios and vertical wind velocity (Lenschow et al., 1981; Anctil et al., 1994; Fairaill et al., 2000), but the wind velocity retrieval is complicated by the contamination due to platform motion, representing a major source of uncertainty in measurement of turbulence and air-sea interaction. Several techniques have been used to correct the wind vector measured at sea for the influence of platform motion (Fujitani, 1992; Dunckel et al., 1974; Song et al., 1996; Edson et al., 1998; Schulz et al., 2005). Fujitani (1985) used a Stable Platform System (SPS) consisting of a vertical gyro-stabilized system and three accelerometers to measure the turbulent flux on the ship, and concluded that this system was applicable to measurement under rough sea surface condition. A similar method was also used on a buoy (Anctil et al., 1994). This gyro-stabilized system can provide roll, pitch and yaw angles describing the ship's orientation in a fixed frame, which can be used directly in the total rotational coordinate transformation matrix. Song et al. (1996) used a Strapped-Down System (SDS) consisting of six accelerometers to measure the air-sea fluxes in the Western Tropical Pacific and estimated that the system appeared to be relatively robust for use at sea for extended period. In SDS, the attitude angles are calculated indirectly from the strapped-down angular rate sensors. Edson (1998) also used the SDS consisting of three orthogonal angle rate sensors and three orthogonal linear accelerometers to compute direct covariance fluxes from anemometers mounted on a moving platform at sea, and found that the results were in good agreement with fluxes derived using the bulk aerodynamic method. Miller et al. (2008) modified the procedure of Edson (1998) procedure to explicitly account for misalignment between anemometers and motion sensors.

The Coherent Doppler Lidar (CDL) has proven to be powerful tool with high temporal and spatial resolution, providing nearly continuous particle backscatter and wind profile observation in the cloud-free atmosphere, which is vital for the vertical structure of turbulent characteristics measurement in MABL. Unlike the conventional in-situ wind measuring methods, CDL can only detect the line-of-sight (LOS) velocity which is the projection of the horizontal and vertical velocity along the laser beam direction, thus it is necessary to conduct measurement at three or more different directions of the probing beam to retrieve the wind vector (Werner 2005; Cheong et al., 2008). More complicated attitude correction need to be considered when CDL is carried out at a moving platform such as ship or aircraft since the orientation of transmitting laser beam is not fixed and the speed of the ship itself and ocean wave will be stacked to the LOS velocity, which has a more serious detrimental effect on vertical velocity. Several studies have been carried out to investigate the CDL platform motion correction either by actively stabilizing the instrument based on robust mechanical compensation system or by accurately measuring platform motion and correcting for this in post-processing. Wolfe et al. (2007) and Pichugina et al. (2012) deployed the NOAA High Resolution Doppler Lidar (HRDL) along with the first use of a motion compensation system at sea in 2004. The HRDL control computer can drive the scanner to actively stabilize the pointing of the scanner and modified Velocity Azimuth Display (VAD) technique are used in the mean-profile calculation. Hill et al. (2005; 2008) used the NOAA HRDL with a SDS to compensate for the orientation of the lidar's scanning unit for the ship's motion and concluded that the attitude correction depends on the velocity

of the seatainer and on the motion of the hemispheric scanner relative to the seatainer. Wulfmeyer et al. (2005) corrected the vertical velocity using LOS velocity in zenith stare mode and horizontal wind derived from VAD mode using NOAA HRDL. Lacking real-time control of the scanning head orientation, Achtert et al. (2015) placed a CDL instrument on a motion-stabilization platform to remove the effect of ship motion, and the five-point geometrical wind solution and the four-point

sinusoidal fit method were used to obtain wind profiles, showing that motion stabilization was successful for high wind speed in open water and the resulting wave condition. Reitebuch et al. (2001) presented the instrumental correction required for the horizontal wind retrieval from an airborne CDL using conical scanning pattern measurement and recalculation of the lidar mounting angle based on the ground return speed and distance. It can be seen that actively mechanical compensation system is used in most of these studies. Especially, the improvements in technology along with decreasing costs and robust correction

process are increasing needed. In order to simplify the mechanical structure and to ease installation of the CDL on the ship platform, an algorithm-based attitude correction method was developed for relaxing the requirements for mechanical stability and active compensation mechanisms. This method did not use any active stabilization method. Instead, only a relative simple but robust algorithm was used to achieve the motion correction in post-processing, which is very easy to use in limited space under the conditions of the shipborne measurements.

The experimental investigation was undertaken by *Dongfanghong-2* research vessel affiliated with Ocean University of China in 2014 over the Yellow Sea. The Yellow Sea, a marginal sea of the Pacific Ocean, is the northern part of the East China Sea. It is located between mainland China and the Korean Peninsula. There is seldom study on boundary layer dynamics study based on CDL in this region. As one of the main objectives, the CDL was deployed on the ship in this campaign to demonstrate the feasibility of the algorithm-based attitude correction method. The obtained accurate three-dimensional wind information

can provide significant preparation for further studies on characteristics of dynamics and thermodynamics in the MABL and turbulence flux exchange over sea surface. In addition to CDL, as another important part of this campaign, a High Spectral Resolution Lidar (HSRL) and a Vaisala CL31 ceilometer were also deployed on the ship platform in order to detect the spatial-temporal evolution of the MABL height and to retrieve the aerosol and cloud optical characteristics such as extinction coefficient and backscatter ratio and so forth. It will help us to understand the complex behaviour of MABL and the aerosol

cloud forcing characteristics over sea region and the impact on climate change. This paper focuses on CDL performance and gives a thorough analysis of the attitude correction for lidar velocity measurement. To illustrate the effect of ship motion on Doppler measurement, we focus on horizontal and vertical wind profile analysis. In Sect. 2, the specifications of CDL and especially its attitude correction system are described in detail, and the velocity correction method is discussed in. In Sect. 3, the corrected results of horizontal wind are analysed and compared with simultaneous radiosonde data. A case study is

presented to analyse the effect of the ship velocity and horizontal wind on vertical velocity. Furthermore, the errors of horizontal and vertical velocity are analysed in Sect. 3. Finally, Sect. 4 provides a summary and concluding remark.

## 2 Lidar technology and methodology

The CDL system WindPrint S4000, manufactured by Seaglet Environmental Technology, is based on all-fibre laser technology and the heterodyne detection technology. Wu et al. (2016) and Zhai et al. (2017) provide a comprehensive description of the CDL including a figure of the optical setup. The lidar has a semi-conductor single frequency seed laser that provides both the local oscillator reference beam for heterodyne detection as well as the transmitted beam. The laser operates at a wavelength of 1.55 μm with a linewidth (full width at half maximum of a Lorentzian function) of 1.6 kHz . Using the Acoustic Optic Modulator (AOM) and Master Oscillator Power-Amplifier (MOPA) configuration, the achieved pulsed energy is approximately 150 μJ and the pulse repetition frequency is 10 kHz . The number of laser shots used for each spectrum accumulation is 5000. The pulse width produced by the modulation, which is also the width of time window for obtaining the lidar signal power spectrum, is adjustable from 100 ns to 400 ns , thus the spatial resolution can be varied from 15 m to 60 m . We typically operate the CDL with a pulse width of 200 ns for this study. The transmitted beam is directed into the atmosphere using the 3D scanner that contains an azimuth and elevation mirror. The scanner allows the lidar beam to probe the hemisphere above the container by means of the "azimuth rotation" and "elevation rotation". The detection range of 4000 m (maximum 6000 m for high aerosol concentration) enables the system to monitor the complete MABL structure most of the time. A fibre optical circulator and a telescope are used as the optical transceiver. The atmospheric return beam passes through the 3D hemispheric scanner and the optical transceiver, and is combined with the local oscillator reference beam at the balanced detector. Using heterodyne detection, the frequency difference between the atmospheric return beam and the local oscillator reference beam is detected, which is the measured Doppler shift caused by the relative motion of atmospheric scatters and the lidar system. The real-time analysis based on Fast Fourier Transform (FFT) is used with a Field Programmable Gate Array (FPGA) signal processer. Table 1 lists the general specifications of CDL.

Figure 1 shows the CDL setup on Dongfanghong-2 research vessel during the MABL field project over the Yellow Sea in 2014. It is located at the back of the upper deck and is around 8 m from the sea surface. The CDL scanner is mounted on the roof of the cabinet container with two fixed Global Navigation Satellite System (GNSS) antennas. Double antennas are used for determining the exact heading angle with accuracy of $0.1°$ when the ship is anchored. The attitude correction system uses XW-GI5651 Micro-electromechanical Systems (MEMS) Inertial/Satellite Integrated Navigation System. It is equipped with MEMS gyroscope, accelerometer, and multi-mode and multi-frequency GNSS receiver. It can realize single antenna dynamic alignment or double antenna auxiliary fast and high-precision orientation. The specifications are listed in Table 2. Generally, the attitude correction system uses GNSS to define Earth coordinate system (ECS), where the ship speed, heading angle and earth location including the longitude and latitude in ECS can be obtained. Another important part of attitude correction system is the inertial navigation system (INS). The INS is rigidly mounted on the base of the scanner within the cabinet container, instead of the deck of the ship, to keep constant relative angles with reference to the lidar coordinate system. It records the lidar motion angles in real time including pitch, roll, laser beam azimuth and elevation even when the GNSS is sheltered or

disturbed, and the recorded information is the exact lidar itself attitude in lidar coordinate system. After installation, a hard target calibration is firstly performed to determine the misalignment between the ship and laser beam axes. Specifically, the buildings near the wharf where there is no occlusion issue between the CDL and the candidate buildings can be chosen as the hard target. The distance between hard target and lidar is about 300 m in this experiment. As shown in Fig. 2, the ECS, ship

coordinate system (SCS) and lidar coordinate system (LCS) are marked with black, red and green arrowed lines, respectively. When the laser beam direction points to the hard target, the azimuth angle $\varphi_{lidar}$ in LCS is recorded, meanwhile the azimuth angle $\varphi_g$ in ECS can be obtained using the Google Earth software if the exact longitude and latitude of hard target is determined. According to the ship heading angle $\psi$, we can get the azimuth angle $\varphi_s = \varphi_g - \psi$ between ship heading and the hard target in SCS. So far, the misalignment angle between the ship and laser beam axes $\Delta\varphi = \varphi_s - \varphi_{Lidar}$ can be corrected

using the geometrical relationship between these three angles. And then the standard ship attitude definition can be determined based on the relationship between lidar and ship coordinate system, which will be used in the following ship motion correction process. It can be seen that the laser direction error determined by misalignment between the ship and Lidar is negligible since the Lidar is considered to be relative static compared with ship during the field experiment. Ship motion turns out to be an important error source for the determination of turbulence variables using shipborne CDL (Wulfmeyer and Janjic, 2005). To

study boundary layer dynamics, the atmospheric wind velocity in ECS is required, so the compensation for the pointing error and along-beam platform velocity due to ship motion need to be determined using the attitude correction system. Figure 2 shows the specific definition of the parameters in SCS ( $X_s, Y_s, Z_s$ ), ECS ( $X_g, Y_g, Z_g$ ) and LCS ( $X_{Lidar}, Y_{Lidar}, Z_{Lidar}$ ), respectively. Details on the motion-correction algorithm are given in Appendix A.

For a shipborne CDL, the recorded velocity corresponds to the relative velocity along the laser beam direction between the

ship and the atmospheric target, where the ship platform motion will add to the measured LOS velocity in SCS. Therefore, the first step in the wind retrieval process is the removal of the along-beam platform velocity due to ship motion $\vec{V}_{LOS\_ship}$. It is noted that wave-induced velocity perturbations would add to the ship's mean velocity when underway, which needs no correction independently in the correction procedure. During the experiment, the speed of the ship $\vec{V}_{ship}$ is acquired by GNSS with temporal resolution of 0.5 s, and is recorded as the horizontal component $\vec{V}_{ship\_horizontal}$ and vertical component $\vec{V}_{ship\_vertical}$,

respectively, thus the $\vec{V}_{LOS\_ship}$ can be calculated

$$\vec{V}_{LOS\_ship} = \vec{r}_g \cdot \vec{V}_{ship} = \vec{V}_{ship\_horizontal} \cos(\psi - \varphi_g) \cos\theta_g + \vec{V}_{ship\_vertical} \sin\theta_g \tag{1}$$

The LOS velocity $\vec{V}_{LOS}$ in ECS is the vector sum of the LOS velocity measured by CDL in SCS $\vec{V}_{LOS\_measure}$ and the $\vec{V}_{LOS\_ship}$, that is,

$$\vec{V}_{LOS} = \vec{V}_{LOS\_measure} + \vec{V}_{LOS\_ship} \tag{2}$$

and

$$\vec{V}_{LOS} = \vec{r_g} \cdot \vec{V} + \vec{r_g} \cdot \vec{W} = u\cos\varphi_g \cos\theta_g + v\sin\varphi_g \cos\theta_g + w\sin\theta_g \tag{3}$$

where $\vec{V} = [u, v, 0]$ and $\vec{W} = [0, 0, w]$ are the horizontal and vertical component of the wind speed respectively, $u$ $v$ and $w$ are the north-south, east-west and vertical velocity in ECS, respectively. The CDL measured LOS velocity has the same temporal resolution of 0.5 s as the $\vec{V}_{ship}$ parameters acquired by GNSS.

Profiles of the wind vector can be retrieved by scanning the lidar beam or stepping the lidar beam through a sequence of
different angles or perspectives (Reitebuch et al., 2001; Frehlich, 2001; Werner 2005). For the ground-based CDL, the profile of horizontal wind velocity can be retrieved using 4-Doppler Beam Swing (DBS) mode which is faster and simpler both in the hardware and in the data evaluation algorithm (Werner 2005; Weitkamp, 2005; Wang et al., 2010). Specifically, the wind vector components at target altitude can be derived by measuring the LOS wind velocities in four directions (normally east, west, south and north) under the assumption of homogenous flow with little turbulence. But for the shipborne platform, the
elevation $\theta_g$ in four directions (north, south, west and east in ship coordination system) may have slightly difference (see Eq. (A5)) due to ship rotation and movement during the time period of measuring different LOS directions. A conversion of $\vec{V}_{LOS}$ from real elevation $\theta_g$ to the expected elevation $\theta_0$ is firstly processed, that is,

$$\vec{V}'_{LOS} = \vec{V}_{LOS}\cos\theta_0 / \cos\theta_g \tag{4}$$

In this study $\theta_0 = 60°$ is set for horizontal wind profile retrieval. During the experiment, each radial direction will take 5 s to obtain 10 measured LOS velocity for accumulation and average. In this sense, the highest temporal resolution of horizontal
wind velocity using 4-DBS mode is 20 s. The recorded ship condition information has the same update rate of 0.5 s as radial velocity's, which can be averaged to remove the platform motion effect on radial velocity.

Furthermore, since the laser beam azimuth angle in ECS need to be determined using Eq. (A4), the conventional DBS formula where four directions at an interval of azimuth-angle of $90°$ are detected need to be modified. Except for the extremely rough sea condition, the LOS velocity component from vertical velocity in different directions is assumed to be identical. Then the
$u$, $v$ can be calculated using a modified 4-DBS formula

$$\begin{pmatrix} \cos\varphi_N - \cos\varphi_S & \sin\varphi_N - \sin\varphi_S \\ \cos\varphi_E - \cos\varphi_W & \sin\varphi_E - \sin\varphi_W \end{pmatrix}\begin{pmatrix} u \\ v \end{pmatrix} = \begin{pmatrix} (\vec{V}'_{LOS\_N} - \vec{V}'_{LOS\_S})/\cos\theta_0 \\ (\vec{V}'_{LOS\_E} - \vec{V}'_{LOS\_W})/\cos\theta_0 \end{pmatrix} \tag{5}$$

$$u = \frac{a_4 b_1 - a_2 b_2}{a_1 a_4 - a_2 a_3} \tag{6}$$

$$v = \frac{a_1 b_2 - a_3 b_1}{a_1 a_4 - a_2 a_3}$$
(7)

where $a_1 = \cos\varphi_N - \cos\varphi_S$ , $a_2 = \sin\varphi_N - \sin\varphi_S$ , $a_3 = \cos\varphi_E - \cos\varphi_W$ , $a_4 = \sin\varphi_E - \sin\varphi_W$ , $b_1 = (\vec{V}'_{LOS\_N} - \vec{V}'_{LOS\_S})/\cos\theta_0$ , and the subscript $N$ , $S$ , $E$ , and $W$ represent the north, south, east and west in SCS , respectively. It is noted that under extremely rough sea condition, the difference of elevation angle in different directions is significant, and the contribution of vertical velocity to LOS velocity needed to be treated carefully. In this case, the height interpolation of radial velocity can be

used, and if three or more radial velocities at the same height are obtained, the horizontal and vertical velocity can be retrieved. But if the elevation angle in one direction is too small, the detectable height will be limited.

For the case of vertical wind measurement, small deviation from vertical pointing due to ship motion induces a projection of the horizontal wind on the laser beam direction. To exactly correct this effect, estimation of the horizontal wind using Eqs. (5), (6) and (7) are used and then the vertical velocity $w$ can be obtained using Eq. (3), where in this formula $\vec{V}_{LOS}$ is the

measurement in zenith stare mode in ship coordination system.

The sampling rate in our study is $f = 1$ GHz , corresponding to the sampling interval of $T_s = 1$ ns . By applying the discrete Fourier transform to $M = 256$ samples, the spectrum estimate has a frequency resolution of $\Delta f = (MT_s)^{-1} = 3.906$ MHz and a velocity resolution of $\Delta V = (\lambda/2)\Delta f = 3.027$ ms$^{-1}$ , where $\lambda$=1.55 μm is the wavelength. In order to obtain the spectrum with better resolution, the interpolation method of adding 768 zeros to the 256 selected samples of the backscatter signal is

used, thus the final frequency resolution is $\Delta f \approx 0.98$ MHz and the corresponding velocity resolution is $\Delta V = 0.76$ ms$^{-1}$ .

Based on the heterodyne detection, the frequency of the local oscillator $f_0$ is optically mixed with the backscatter signal $f_0 + f_{IF} + \Delta f_D$ , where $f_{IF}$ is the intermediate frequency and $\Delta f_D$ is the Doppler shift from atmospheric movement. Thus the beat signal of $f_{IF} + \Delta f_D$ is obtained in a detectable range $B$ of MHz . Figure 3 shows the array of the FFT spectrum $S(l\Delta f; k\Delta R)$ after zero padding interpolation obtained from the raw data measured with PCDL, where $l = 0,1,2,3,...,L-1$ is

the spectral channel number and $L = 100$ and $k = 1,2,3...,K$ is the range bin number and $K = 104$ . It is noted that the 100 spectral channel is symmetrically selected near the intermediate frequency with $B_{100} = (L-1)\Delta f = 97.68$ and corresponding radial velocity measurement range of $\pm37.5$ ms$^{-1}$ .

The scattering particles in the sensing volume have a certain velocity distribution. It is easy to estimate the mean radial velocity from the FFT spectrum estimate with high SNR. However, the evaluation of the maximal velocity with acceptable accuracy is

not possible because of the strong fluctuations of the signal and noise components in the spectrum estimate. In order to reduce these fluctuations, the spectrum accumulation is used. In our study the pulse repetition rate is 10 kHz and the accumulation shot is 5000 for each radial velocity measurement.

The SNR in this study is defined as the ratio of the peak value of FFT spectrum signal in each range bin to the Root-Mean-Square (RMS) of background noise signal. Figure 3a shows the last 10 range bins' raw array of spectrum in green line. We estimate the averaged background noise spectrum

$$\overline{S}_N(l\Delta f) = \frac{1}{10}\sum_{k=94}^{103} S(l\Delta f; k\Delta R) \tag{8}$$

Subtracting the background noise spectrum $\overline{S}_N(l\Delta f)$ from the raw spectrum array $S(l\Delta f; k\Delta R)$, the unnoisy array of

spectrum $S(l\Delta f; k\Delta R)$ can be obtained and shown in red line in Fig. 3. The peak value index $l_{peak}$ from the $S(l\Delta f; k\Delta R)$

can be firstly obtained and thus the absolute signal power $P_s(k\Delta R)$ at various range $k\Delta R$ can be represented as:

$$P_s(k\Delta R) = S(l_{peak}\Delta f; k\Delta R) - \frac{1}{12}\left(\sum_{l_{peak}-20}^{l_{peak}-15} S(l\Delta f; k\Delta R) + \sum_{l_{peak}+15}^{l_{peak}+20} S(l\Delta f; k\Delta R)\right) \tag{9}$$

Replacing integration by summation and taking into account that the zero velocity point in one channel is $l_{zero} = 50$, we estimate the noise power $P_N$ as

$$P_N = \frac{1}{10}\sum_{k=94}^{k=103}\sqrt{\frac{1}{21}\sum_{l=l_{zero}-10}^{l_{zero}+10} \hat{S}_N(l\Delta f; k\Delta R)^2} \tag{10}$$

Finally, we obtain the range profile of the $SNR(k\Delta R)$ using the equation

$$SNR(k\Delta R) = 10\log_{10}\left(\frac{P_s(k\Delta R)}{P_N}\right) \tag{11}$$

It is noted that unlike the definition of SNR in previous studies (Banakh et al. 2013) where the SNR is defined as the ratio of the average heterodyne signal power to the averaged detector noise power in a 50-MHz bandwidth, the SNR in this study is simpler and also indicates the CDL detection capability, data accuracy and atmospheric tracer particle relative intensity. In this sense, the SNR threshold value in this study is higher than the one in previous studies (Banakh et al., 2013; Achtert et al., 2015) for the same signal power spectrum.

Figure 4 shows the flowchart of shipborne CDL data processing. Specifically, the LOS velocity and SNR can be firstly determined using lidar data and FFT analysis. After the data pre-processing including the quality control based on SNR threshold, the attitude transformation is then used to obtain the azimuth and elevation in each LOS velocity in ECS with temporal resolution of 0.5 s. The LOS velocity detected by the lidar is the atmosphere motion relative to SCS, thus the removal of the along-beam platform velocity due to ship motion is needed. In this study, the horizontal wind profile with 2 minutes

temporal resolution will be retrieved for vertical velocity correction. Basically, the LOS velocities from $N$, $S$, $E$, and $W$ direction after SNR quality control during the chosen 2 minutes interval are collected firstly. Then the procedure of filtration of reliable estimates of each radial velocity based on SNR threshold is used to obtain "good" speed estimates. The selected

radial velocities and corresponding ship condition information in each radial direction are averaged and the averaged ship condition will be used for the removal of platform velocity effect. Finally, the horizontal velocity with 2 minutes temporal resolution can be retrieved using modified 4-DBS mode. The vertical wind measurement has a temporal resolution of 0.5 s, the horizontal wind whose retrieved time is closest to vertical wind measured time will be used for vertical velocity correction.

## 3 Observation results and discussion

### 3.1 Horizontal wind evaluation

The modified 4-DBS method for horizontal wind profile retrieval is illustrated in Fig. 4. Two examples of the comparison between uncorrected and corrected horizontal wind profiles are shown in Fig. 5 for anchored measurement and Fig. 6 for cruising observation, respectively. The temporal resolution of radial velocity is 0.5 s and the lidar results (black curves) averaged over at least 10 min after the launch of the radiosonde are compared with the radiosonde data (red curves). The type of radiosonde is Model GTS1 digital radiosonde with the basic parameters listed in Table 3 (Song et al., 2017).

Figure 5 shows the horizontal wind profile during anchored measurement (mean ship speed equals to 0.27 $\mathrm{ms}^{-1}$) during 15:52-16:02 Local Standard Time (LST) on 9 May 2014 at $37.00^{\circ}$ N, $122.86^{\circ}$ E. The black line in Fig. 5b – Fig. 5e indicates the mean measurement by CDL during the 10-min period, and the red line shows the result from simultaneous radiosonde data. The blue bars represent the standard deviation of CDL wind measurement from the 2 minutes temporal resolution results during the chosen analysed period, which can effectively represent the atmospheric fluctuations. It can be seen that the wind is approximately southerly through the measurement altitude, but slightly southeasterly below 1.6 km, and then shifts to southwesterly above 1.6 km. The wind speed gradually decreases with height till 1.6 km and then increases above. The standard deviation of wind speed and direction below 1.4 km are less than 0.5 $\mathrm{ms}^{-1}$ and $5^{\circ}$, respectively, showing that the atmospheric condition is relative stable below 1.4 km. While there are higher fluctuations in the height of 1.4 – 1.6 km. The higher SNR in the layer of 1.4 – 1.6 km shown in Fig. 5a implies the existence of a cloud or aerosol layer, more active and complex atmospheric movement in this layer may result in higher fluctuations. The specific ship condition parameters are listed in Table 4. The time series of ship horizontal speed, pitch and roll angles are shown in Fig. 7a and 7b, respectively. It can be seen that the mean pitch and roll are $-0.17^{\circ}$ and $0.63^{\circ}$ with standard deviation $0.06^{\circ}$ and $0.11^{\circ}$, respectively, thus the swing of the ship is not obvious. It is noted that the standard deviation of the angles is determined from the variability during the 10 min period using N=1200 raw data with temporal resolution of 0.5 s, which is shown in Fig. 7b. Since lower SNR makes the data invalid, data quality control based on SNR threshold is used to remove the spikes higher than 2.4 km. The SNR threshold in this study is 8 dB and the reason will be analysed in Sect. 3.3. The Root- Mean-Square-Error (RMSE) in speed between lidar and radiosonde below 2 km is 0.49 $\mathrm{ms}^{-1}$ for the uncorrected measurement and 0.45 $\mathrm{ms}^{-1}$ for the corrected measurements, both showing consistent with the radiosonde wind speed. It is reasonable since the effect of ship motion speed on LOS velocity is less obvious in anchored measurement. Moreover, the variation of lidar elevation and azimuth in ECS is

small, and in this case, for instance, when the lidar points to bow with elevation of $60°$ in SCS. If the ship's pitch, roll and heading are $-0.17°$, $0.63°$, $5.28°$, respectively, according Eqs. (A4) and (A5), the lidar azimuth and elevation in Earth coordinate are $\varphi_s = 6.37°$ and $\theta_s = 59.82°$, respectively. Similarly, when the lidar points to starboard, stern and port, the corresponding azimuth are $94.99°$, $184.18°$ and $275.58°$, and the elevation are $59.37°$, $60.16°$ and $60.63°$, respectively, resulting in less difference of horizontal wind speed retrieved from the SCS and ECS. However, the RMSE in wind direction between lidar and radiosonde is $84.43°$ for uncorrected measurement and $5.27°$ for corrected measurement. The obvious difference of the wind direction results from two aspects. The first one is the definition in different coordinate systems, where the heading has an important effect on lidar azimuth. The second aspect is that because of the experimental field limitation, the direction of GNSS master antenna is perpendicular to the ship bow, meaning that the "real" heading is the recorded heading plus $90°$, and this angle offset due to placement problem is fixed and calibrated using hard target detection before the campaign. Generally, attitude correction is necessary, especially for the wind direction retrieval even though the ship is anchored with slight shake.

Figure 6 shows the results of the cruising observation from 07:44 to 07:54 LST on 13 May 2014 when the mean ship speed is $4.84$ $ms^{-1}$ with standard deviation of $0.03$ $ms^{-1}$. It can be seen that the wind is constantly southwesterly through the available measurement altitude, and there is a low-level-jet at around $0.3$ $km$ where the wind speed exceeds $25$ $ms^{-1}$. What's more, the fluctuation in wind speed and direction above 1 km is more severe than the result below 1 km. The specific ship condition parameters are also listed in Table 4. The time series of ship horizontal speed, pitch and roll angles are shown in Fig. 7c and 7d, respectively. It can be seen that the mean pitch and roll are $-0.43°$ and $2.06°$ with standard deviation $0.05°$ and $0.87°$, respectively. Generally, the ship roll has a more effect on the lidar elevation when it points to the port or starboard, on the contrary, the lidar elevation in bow or stern direction is more sensitive to ship pitch. In this case, the lidar mean elevation in bow, starboard, stern and port direction after attitude transformation are $59.51°$, $57.84°$, $60.30°$ and $62.49°$, respectively, and the mean heading is $75.86$ with standard deviation $1.22°$ where the ship sails downwind. In this condition, the horizontal wind speed without motion correction will be underestimated compared with the radiosonde result. The RMSE in speed between lidar and radiosonde data below $1.0$ $km$ are $4.42$ $ms^{-1}$ for uncorrected measurement and $0.88$ $ms^{-1}$ for corrected measurements, and the corresponding RMSE in wind direction are $48.71°$ and $9.52°$, respectively. Therefore, the attitude correction algorithm has obviously improved the wind profile result when the ship is in cruising observation. The difference in mean wind speed and direction between radiosonde and CDL above 1 km is about $3.4$ $ms^{-1}$ and $15.2°$, respectively, showing significant discrepancy. On the one hand, the random error of the corrected CDL estimation of the wind due to the low SNR shown in Fig. 6a contributes to this discrepancy. On the other hand, according to the recorded information, the mean heading angle and cruising speed of the ship is $75.86°$ and $4.84$ $ms^{-1}$, respectively, and the mean wind speed and direction above 1 km is $255°$ and $18.4$ $ms^{-1}$, respectively. Since the drift of radiosonde is affected by atmospheric wind and turbulence

perturbation, and the CDL detection volume is changing during cruising observation, the result discrepancy between radiosonde and CDL caused by different observation location, also called the multipath effect, is larger with increasing height. In order to assess the accuracy of the shipborne lidar wind measurement, a comparison of the lidar measurement and 11-radiosonde dataset during the experiment are made. It is noted that the radial range resolution of lidar in this study is 30 m,

and the corresponding vertical range resolution with elevation angle of $60°$ is about 26 m. The vertical resolution of radiosonde is 10 m. During the comparison, the wind profile of radiosonde is interpolated into the common height grid with finer resolution of 2 m firstly, and then the data point closest to height point of lidar will be chosen for comparison. Figure 8 shows a scatter plot of wind speed and direction for radiosonde and lidar measurement based on modified 4-DBS solution. The radial measurement range in this study is between 150 m and 3240 m (corresponding to the 104th range bin), thus the altitude range

between 130 m and 2806 m with elevation angle of $60°$ are used for the statistical comparison shown in the Fig. 8. The red trend line plotted through these points represents an ordinary linear least square regression for the data excluding $|ydata - xdata| > 2 \times SD$, where $ydata$ and $xdata$ is the lidar and corresponding radiosonde data, respectively, and SD represents the standard deviation of the difference of $ydata - xdata$. According to the distribution of difference of $ydata - xdata$ and fitted Gaussian distribution, the criteria of excluding data with $2 \times SD$ is reasonable for gross outliers. The

excluded data-pair number and proportion is 62 and 6% for wind speed, respectively, 56 and 5.9% for wind direction, respectively. The wind speed linear regression shows a correlation coefficient of 0.982, SD of 0.87 $ms^{-1}$ and RMSE of 0.90 $ms^{-1}$. The wind direction linear regression shows the correlation coefficient of 0.995, SD of $8.84°$ and RMSE of $9.50°$. The bias of wind speed and direction is -0.23 $ms^{-1}$ and $2.48°$, respectively, demonstrating the feasibility and reliability of the modified 4-DBS solution.

Table 5 lists a height-resolved view (from 0.2 $km$ to 1.6 $km$) of the linear fit parameters between lidar and radiosonde. The correlation coefficient R for wind direction is approximately 0.99 and almost constant with altitude up to 1.6 $km$. The correlation coefficient for wind speed is minimum at the lowest altitudes, and improves with height to values comparable to those for wind direction, the trends of which compare well with the results from Achtert et al. (2015). An obvious feature in SD, RMSE, normalized RMSE for wind speed and direction is found at the lowest levels where the discrepancies between

lidar and radiosonde data are larger than the higher levels. On the one hand, the relative height between CDL and the highest building on ship is about 15 m shown in Fig. 1b. When the strong wind blows from the ship bow, the building and experimental setups on ship have an important effect on CDL lower-level detection volume where the induced-turbulence may cannot meet the assumption of homogeneous isotropic atmosphere for 4-DBS retrieval. On the other hand, the blind area of the CDL is 150 m and corresponds to the height of 129.9 m when laser beam elevation angle is $60°$, meaning that less data points are available

below 200 m with effective comparison. Whether the flow distortion around the ship is the main reason for the discrepancies in the lower part measurement or not is yet unclear. Further study, especially focused on the Computational Fluid Dynamics

(CFD) model, needs to be used to the assess the potential effects on turbulent flow and wind field analysis (Achtert et al., 2015). It is also significant for assessing turbulence fluxes exchange from sea-air interface. Furthermore, SD, RMSE, normalized RMSE for wind speed and direction increase with altitude from 0.4 km, which are mainly caused by the lower SNR and increasing spatial separation because of the multipath effect mentioned above.

## 3.2 Vertical wind evaluation

The motion correction of vertical velocity, which is more challenging compared with the horizontal wind component, has been specifically described in Sect. 2. A typical measurement case study on 14 May 2014 is presented in Fig. 9. Figure 9a shows the whole series of time-height cross section of the SNR. According to the Vaisala CL31 ceilometer recorded result, an aerosol layer is presented at around 2.1 km during 07:33 – 08:40 LST. The MABL height has been retrieved and compared using different instruments such as the CDL, radiosonde, and Vaisala CL31 ceilometer during this campaign (Wang et al., 2016). Many papers have discussed the use of backscatter signal of Lidar for mixing height estimation, assuming that the boundary layer has higher aerosol concentrations than the free troposphere above. In this study, the SNR, representing the relative aerosol backscatter profiles, were used and two common methods includes thresholding SNR to determine MABL height (Melfi et al., 1985) and finding the height of the first strong negative gradient (White et al., 1999; Hennemuth and Lammert, 2005) in SNR. The temporal and spatial variation of MABL height from threshold and gradient methods can be seen in Fig. 9a marked with black and red solid circles, respectively. The radiosonde data during 12:00 LST on 14 May 2014 and corresponding MABL height using the gradient of potential temperature and relative humidity are also shown in Fig. 10. It can be seen that diurnal variation of MABL height is less obvious within 1.0 km - 1.5 km, consistent with the mixing layer height retrieved from the radiosonde potential temperature and relative humidity profile. The corrected vertical velocity wind speed is presented in Fig. 9b. It is noted that the data analysis below 0.15 km is not reliable because of the lidar blind area, and the data above 1.5 km is also removed since the SNR is less than its threshold value. The red and blue colour indicate positive (upward) and negative (downward) movement of the atmosphere parcels along the laser beam, respectively. It can be seen that the vertical velocity has a significant diurnal variation. Specifically, the downdraft dominants mixing layer in the morning and amounted to about 0.5 $ms^{-1}$, and small-scale convective activity can be observed at the top of mixing layer. As the solar radiation strengthens, the atmospheric convection becomes more active and extends to the whole mixing layer, the strengths of updrafts and downdrafts are weakly stronger than before and the atmospheric vertical alternation becomes more frequent. The mixing layer recovers to descending motions with a continuous and long period after 13:11 LST.

Figure 11a shows the time series of ship heading, CDL laser beam azimuth and elevation, and horizontal wind direction at 0.4 km, respectively. Figure 11b shows the time series of elevation angles in zenith stare mode from SCS and ECS, respectively. It can be seen that the hemispherical scanner maintains the pointing of the lidar beam to zenith stare mode with an averaged elevation of $88.6° \pm 0.35°$ because of the ship motion. During the zenith stare mode, the mean angle between ship heading and the laser azimuth is $66°$ with standard deviation of $7°$, thus the projection of ship velocity on vertical velocity is always

positive, the results of which are shown in Fig. 12a. Furthermore, the estimation of the horizontal wind speed and direction (black line in Fig. 11a) from modified 4-DBS solution is used to remove the horizontal wind speed projection $\vec{r_g} \cdot \vec{V}$ from the relative speed measured by CDL. In this case, the $\vec{r_g} \cdot \vec{V}$ is positive and negative in downwind and headwind, respectively, causing the overestimate and underestimate of the vertical velocity, the effect of which is shown in Fig. 12b. The difference

between the corrected and uncorrected vertical velocity can be shown in Fig. 12c, obviously showing the temporal and spatial variation of the contribution of the ship motion and horizontal wind on the vertical velocity.

### 3.3 Measurement uncertainty and error analysis

Error analysis is useful in assessing the accuracy and precision of the lidar wind measurements (Wang et al., 2010). They also shed light on the potential improvements of this CDL. According to the definition of error for measurement of a random wind

field, the measured velocity is represented as (Frehlich, 2001):

$$\hat{V} = V_{truth} + e_V + bias_V \tag{12}$$

where $V_{truth}$ is the desired or true wind measurement, $e_V$ is the random error with zero mean, representing the precision of wind measurements, and $bias_V$ is the systematic error, representing the accuracy of the wind measurements.

As for radial velocity, for instance, the north radial velocity $\hat{V}_{LOS\_N}$ with azimuth angle $\varphi_N$ and elevation angle $\theta_N$, the measurement can be represented as:

$$\hat{V}_{LOS\_N} = c_1 \bar{u} + c_2 \bar{v} + c_3 \bar{w} + e_N + bias_N \tag{13}$$

where $\bar{h} = [\bar{u}, \bar{v}]$ and $\bar{w}$ are the spatially averaged horizontal and vertical velocity, respectively, $c_1 = \cos\varphi_N \cos\theta_N$, $c_2 = \sin\varphi_N \cos\theta_N$, $c_3 = \sin\theta_N$, $e_N$ and $bias_N$ are the random error and bias of the north radial velocity measurements, respectively.

For shipborne measurement, the ship platform velocity $\vec{V}_{ship}$ produces a large contribution $\vec{V}_{LOS\_ship}$ to the total radial velocity (see 1)). The bias in the radial velocity measurement comes from errors in the knowledge of $\vec{V}_{ship\_horizotnal}$, $\vec{V}_{ship\_vertical}$, $\psi_N$,

$\varphi_N$, $\theta_N$,

$$\begin{aligned} bias_{LOS\_N} = {} & \Delta\vec{V}_{ship\_horizontal} \cos(\psi_N - \varphi_N)\cos\theta_N + \Delta\vec{V}_{ship\_vertical} \sin\theta_N - \\ & \Delta\psi_N \vec{V}_{ship\_horizontal} \cos\theta_N \sin(\psi_N - \varphi_N) + \Delta\varphi_N \vec{V}_{ship\_horizontal} \cos\theta_N \sin(\psi_N - \varphi_N) \\ & + \Delta\theta_N (\vec{V}_{ship\_vertical} \cos\theta_N - \vec{V}_{ship\_horizontal} \cos(\psi_N - \varphi_N)\sin\theta_N) \end{aligned} \tag{14}$$

where $\Delta \vec{V}_{ship\_horizontal}$, $\Delta \vec{V}_{ship\_vertical}$, $\Delta \psi_N$ are the errors in the determination of the ship speed and direction, and equal to 0.1 $ms^{-1}$, 0.1 $ms^{-1}$, $0.1°$, respectively. $\Delta \theta_N$, $\Delta \varphi_N$ are the pointing angle knowledge errors of the north direction lidar beam. In this case, $\Delta \varphi$ and $\Delta \theta$ are related to the servo system, and the scanner pointing accuracy is $0.1°$, thus $\Delta \varphi = \Delta \theta = 0.1°$ in all directions. It is noted that the knowledge error of the ship velocity and lidar pointing angle mentioned above are systematic part and it is assumed that the random error of these quantities is zero, which is reasonable and robust for horizontal wind retrieval. Similarly, to derive the $\vec{V}'_{LOS}$ bias, we take the derivatives of Eq. (4)

$$bias_{LOS\_N'} = bias_{LOS\_N} \cos\theta_0 / \cos\theta_N + \Delta\theta_N \vec{V}_{LOS\_N} \cos\theta_0 \sin\theta / \cos^2\theta_N \qquad (15)$$

Because of the requirement for small bias in the radial velocity measurements, the error in the laser beam direction must be very small and one can assume perfect knowledge of the coefficient $a_i$ (Frehlich, 2001), so the biases of $u$ and $v$ from the radial velocity estimation can be described:

$$bias_u = \frac{a_4 bias_{b_1} - a_2 bias_{b_2}}{a_1 a_4 - a_2 a_3} \qquad (16)$$

$$bias_v = \frac{a_1 bias_{b_2} - a_3 bias_{b_1}}{a_1 a_4 - a_2 a_3} \qquad (17)$$

It can be seen that the dominant source of bias of the horizontal velocity estimates come from the biases of the radial velocity estimates ($bias_N$, $bias_S$, $bias_E$ and $bias_W$), which are determined by the error in the ship velocity $\vec{V}_{ship\_horizontal}$, $\vec{V}_{ship\_vertical}$ and heading angle $\psi$ and lidar pointing knowledge errors $\Delta\varphi$ and $\Delta\theta$ (see Eq. 13).

Various methods of estimating the magnitude of the random error of Doppler Lidar velocity measurements have been introduced (Frehlich, 2001). A method based on the frequency spectrum of the retrieved velocity has been used to determine the random error of vertical wind measurements. A 50 % window overlap factor, a Hamming window is used in order to reduce leakage in the spectra (Chouza et al., 2016). A zero-padding of the missing values were applied to each window for each spectrum calculation to improve the frequency resolution. The constant high-frequency region of vertical velocity spectrum higher than 0.2 Hz, shown in Fig. 13 at height of 1495 m, represents uncorrelated random error contribution, which is departing from the Kolmogorov's -5/3 law. The random error of vertical velocity is estimated as the standard deviation of the measured signal after high-pass filter. Figure 14 shows the error analysis of horizontal and vertical wind during 15:52-16:02 LST on 09 May 2014. The observed SNR is illustrated in Fig. 14a, and there is an aerosol layer at around 1.5 $km$, consistent with the higher value in SNR. The random errors of vertical velocity from the standard deviation of the random noise signal, shown in Fig. 14b, are less than 0.1 $ms^{-1}$ below 1 km with SNR< 8 $dB$, and a peak value appears at around 1.3 $km$ and decreases with altitude till at around 1.5 $km$. Then the random errors increase with altitude as the SNR decrease, and reach about 1.2 $ms^{-1}$

at 2.3 $km$. It is clear that the random error is mainly determined by the SNR. Figure 14c shows the $bias_u$, $bias_v$ and corresponding bias of horizontal wind velocity $bias_h$. The $bias_h$ is less than 0.02 $ms^{-1}$ below 2.5 $km$, which is negligible and consistent with the result shown in Sect. 4.1. According to Eqs. (16) and (17), the dominant source of bias of horizontal wind velocity is mainly from the ship velocity and lidar pointing errors in different direction. In this case, $\vec{V}_{ship\_horizotnal}$ provides the highest contribution on the bias of the radial velocity. The observed random error of the vertical velocity as a function of SNR is presented in Fig. 15, which is retrieved from the frequency spectrum of the retrieved vertical velocity during 07:33 to 15:29 LST on 14 May 2014. It can be seen that in the high SNR region above 8 $dB$, a constant random error range between 0.03 and 0.15 $ms^{-1}$ is found because of the effect of the speckle-induced phase noise (Frehlich, 1997; Frehlich, 2001), which is much smaller than the standard deviation between the mean wind speed derived from lidar and radiosonde of 0.75 $ms^{-1}$ (see Sect. 3.1). At reduced values of the SNR, the errors increase as a result of increasing signal noise, rising to approximately 4 $ms^{-1}$ at an SNR = 0 $dB$. It is confirmed that the choice of a conservative SNR threshold of 8 $dB$ is robust for data quality control process.

## 4 Summary

Shipborne wind observation by a CDL during the 2014 Yellow Sea campaign are carried out to study the structure of the MABL. An algorithm to compensate for error of wind measurement due to the motion of the ship is presented in this paper. The algorithm-based attitude and velocity correction methods greatly relax the requirements for mechanical stability and compensation mechanisms. The attitude correction system of the CDL consists of GNSS and INS to directly measure the speed and the attitude of the ship. According to the transformation matrix from the product of roll, pitch and heading rotation matrix, the azimuth and elevation of the LOS velocity in the ECS can be firstly determined. Then the removal of the along-beam platform velocity due to ship motion is needed to obtain the "real" LOS velocity in the ECS. The horizontal wind profiles can be retrieved by a modified 4-DBS method. For the case of vertical velocity, small deviations from vertical pointing due to ship motion induces a projection of the horizontal wind on the LOS vector, thus estimation of the horizontal wind speed contribution are used to correct the vertical velocity.

In order to assess the accuracy of the shipborne lidar wind measurement, a comparison of the lidar measurement and 11-radiosonde dataset from 09 May 2014 to 19 May 2014 are made. The total number of wind speed and direction dataset for comparison is 1062 and 951, respectively. The comparison of the CDL and radiosonde shows that attitude correction is essential for the wind retrieval in cruising measurement. The correlation coefficients of wind speed and direction are 0.982, 0.995, respectively, both of which show negligible bias and demonstrate the feasibility and reliability of the modified 4-DBS method. A case study of 8-h time series observation on 14 May 2014 is presented to compare uncorrected and corrected vertical velocity, additionally showing the specific temporal and spatial variation of the contributions of ship motion and horizontal wind on vertical velocity.

The bias of horizontal wind velocity is estimated using error propagation analysis and concluded that the dominant source comes from the radial velocity estimates, which are determined by the error in the ship velocity and lidar pointing errors. The random error is estimated based on the frequency spectrum of the retrieved velocity. A case study during 15:52 to 16:02 LST on 09 May 2014 is presented. The radial measurement range is from 0.15 km to 3.105 km where the blind area of CDL is less than 0.15 km and the maximum detectable range is 3.105 km. It is found that the random error of vertical velocity is between 0.03 $ms^{-1}$ and 1.2 $ms^{-1}$ and is mainly determined by the SNR, while the bias was less than 0.02 $ms^{-1}$, which is negligible and consistent with the result of comparison between lidar and radiosonde data. The fundamental random error of the lidar vertical wind obtained from 07:33 to 15:29 LST on 14 May 2014 in all height range is found to be in the range of 0.03 to 0.15 $ms^{-1}$ for SNR above 8 dB, which is much smaller than the standard deviation between the mean wind speed derived from lidar and radiosonde of 0.75 $ms^{-1}$. The choice of a conservative SNR threshold of 8 dB is also confirmed by the error analysis results of vertical velocity. Overall, combining a CDL with attitude correction system and accurate motion correction process as presented here forms a reliable and autonomous set-up that could be placed on mobile platform to provide more detailed, higher spatial and temporal resolution view of three-dimensional wind field information. It will be further validated and improved under different sea conditions using CFD model simulation in further field campaign. More specific studies are being carried out or prepared, including atmospheric turbulence characteristics statistics and multi-scale wind field observation in MABL, wind turbine wake and atmospheric turbulence interaction over offshore wind power field (Wu et al., 2016; Zhai et al., 2017), mass transport and flux analysis in MABL with combination of CDL and Multi-wavelength Polarization Raman Lidar (Wu et al., 2016).

**Appendix A:**

As can be seen in Fig. 2, the SCS ($X_s, Y_s, Z_s$) is defined as $X_s$ axis along centre line of ship, positive toward bow, $Y_s$ axis is perpendicular to $X_s$, and positive toward starboard, $Z_s$ axis is positive toward the bottom. The attitude of the ship can be expressed by roll $\varphi$, pitch $\theta$ and heading angles $\psi$. The $\varphi$, $\theta$ and $\psi$ refer to rotations about $X_s$, $Y_s$, and $Z_s$ axes, respectively. Specifically, positive $\varphi$ is defined when the port is up, and positive $\theta$ is defined when the bow is up. The $\psi$ is defined $0°$ when the bow points to north in ECS. The ECS ($X_g, Y_g, Z_g$) is defined as $X_g$ axis along north-south direction, positive toward to north, $Y_g$ axis is along east-west direction, and positive toward to east, $Z_g$ axis is positive toward the bottom. In the SCS, the recorded azimuth and elevation of the transmitting laser are $\varphi_s$ and $\theta_s$, respectively. $\varphi_s$ is defined as the angle between the projection of transmitting laser path on $X_s - Y_s$ plane and the positive $X_s$ axis. From the top view, $\varphi_s$ increases in a clockwise direction during 4-DBS mode operation. $\theta_s$ is defined as the angle between the laser beam direction and the

$X_s - Y_s$ plane. Therefore, the direction of the transmitting laser in the SCS can be expressed by a unit vector $\vec{r_s}$ as (Hill, 2005; Liu et al., 2010).

$$\vec{r_s} = \begin{pmatrix} x_s \\ y_s \\ z_s \end{pmatrix} = \begin{pmatrix} \cos\theta_s \cos\varphi_s \\ \cos\theta_s \sin\varphi_s \\ -\sin\theta_s \end{pmatrix} \tag{A1}$$

The coordinate transformation from the SCS to that of the ECS is needed. According to the transformation matrix from the product of three rotation matrixes shown in Eq. (A2), the unit vector $\vec{r_g}$ of transmitting laser direction in ECS can be expressed

5 as Eq. (A3)

$$H_2 = \begin{pmatrix} \cos\theta & 0 & -\sin\theta \\ 0 & 1 & 0 \\ \sin\theta & 0 & \cos\theta \end{pmatrix}, H_3 = \begin{pmatrix} \cos\psi & \sin\psi & 0 \\ -\sin\psi & \cos\psi & 0 \\ 0 & 0 & 1 \end{pmatrix}, H_3 = \begin{pmatrix} 1 & 0 & 0 \\ 0 & \cos\varphi & \sin\varphi \\ 0 & -\sin\varphi & \cos\varphi \end{pmatrix}, \tag{A2}$$

$$\vec{r_g} = \begin{pmatrix} x_g \\ y_g \\ z_g \end{pmatrix} = (H_1 H_2 H_3)^{-1} \vec{r_s} \tag{A3}$$

Where $H_1$, $H_2$, $H_3$ are the rotation matrices of roll, pitch and heading, respectively (Hill, 2005).

Once the unit vector $\vec{r_g}$ is calculated from Eqs. (A2) and (A3), the azimuth $\varphi_g$ and elevation $\theta_g$ of LOS observation in ECS can be calculated

$$\varphi_g = \arctan(y_g / x_g) \tag{A4}$$

$$\theta_g = -\arcsin z_g \tag{A5}$$

**Acknowledgements**

We thank very much the reviewers for their time and efforts, thoughtful and very useful comments. This work was partly supported by the National Natural Science Foundation of China under grant 41375016 and 41471309,the National High Technology Research and Development Program of China under grant 2014AA09A511, and the National Key Research and
15 Development Program of China under grant 2016YFC1400904.

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

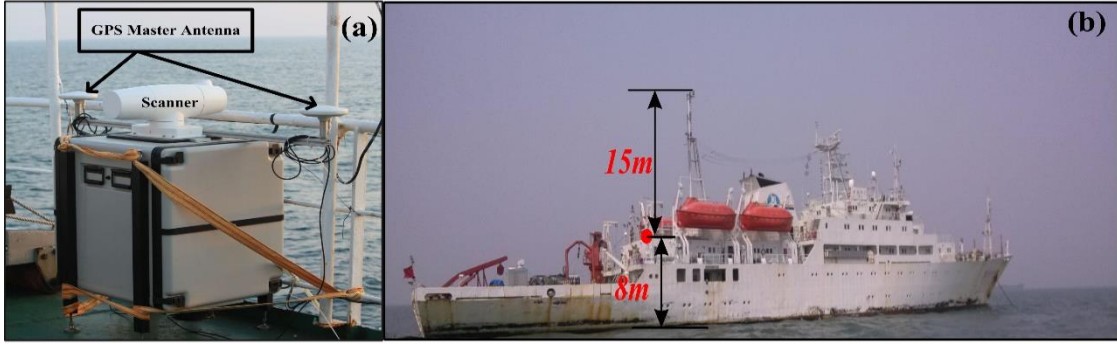

**Figure 1: (a) The Coherent Doppler Lidar setup on Dongfanghong-2 research vessel (b) The Dongfanghong-2 research vessel during 2014 Yellow Sea Campaign. The red solid dot represents the CDL position.**

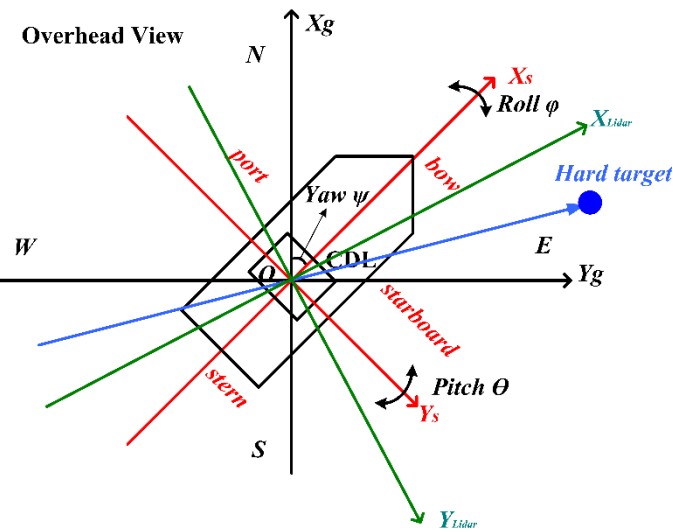

**Figure 2. The overhead view of Lidar, ship and Earth coordinate system and corresponding hard target calibration.**

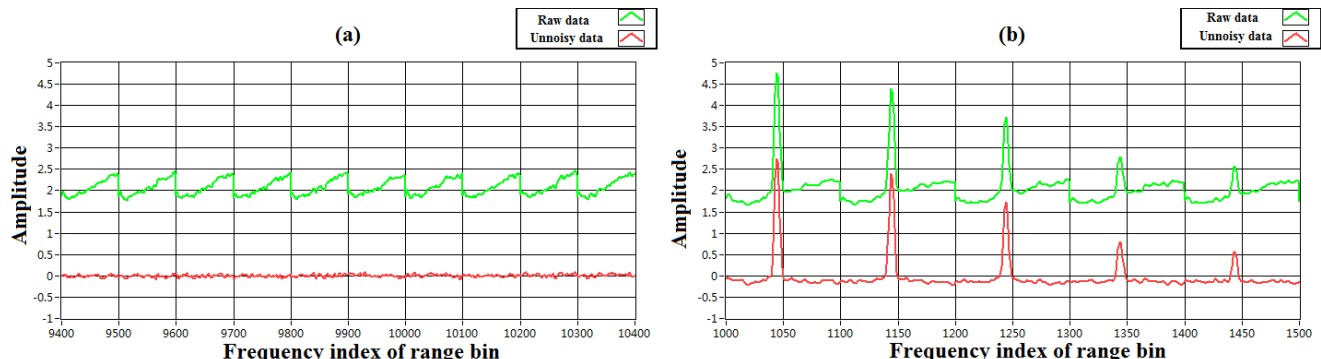

**Figure 3: The CDL measured array of the FFT spectra (a) the last 10 range gates spectra for background noise spectrum estimation (b) the 1st – 5th range gates (150 m – 270 m, range resolution is 30 m) spectrum.**

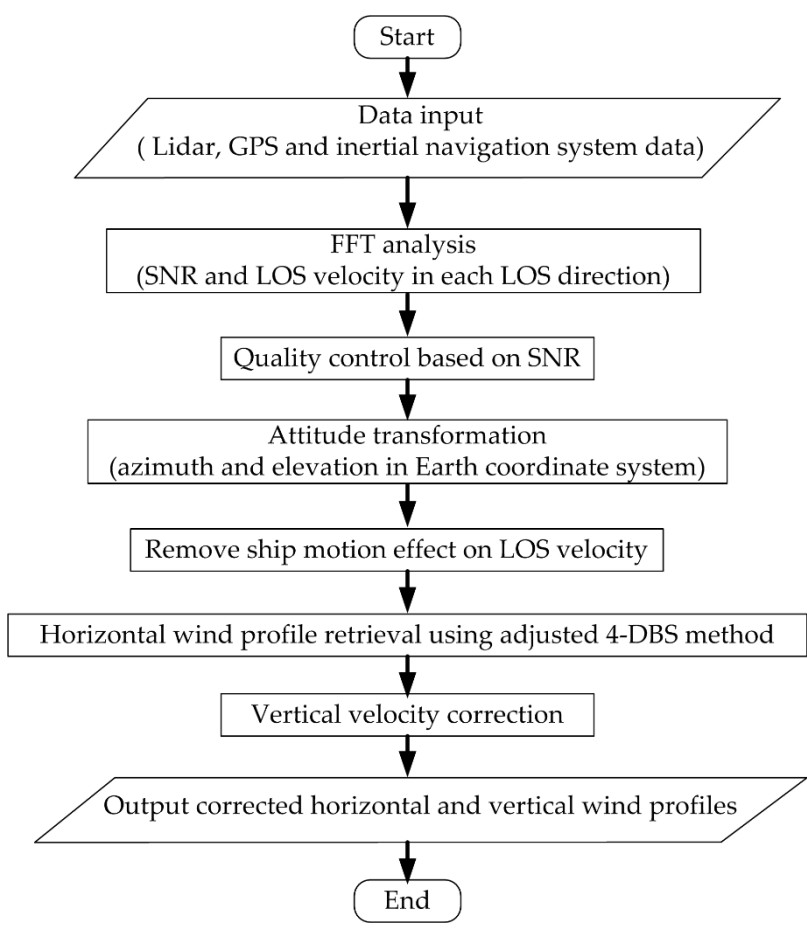

**Figure 4: Flow chart of ship motion correction algorithm based on CDL.**

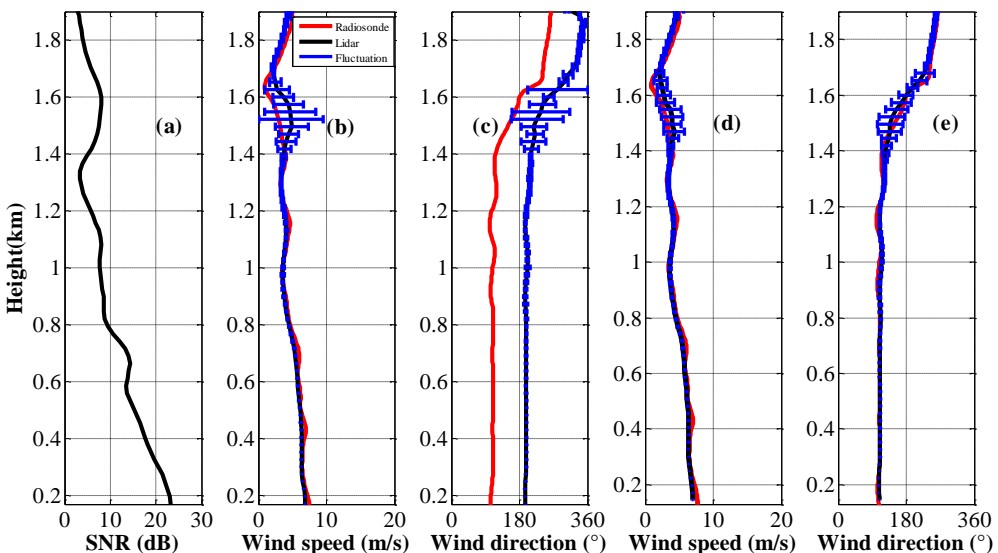

**Figure 5: Anchored observation: (a) SNR profile (b) (c) wind speed and (d) (e) wind direction measured by CDL (black line) before and after attitude correction, respectively. The simultaneous radiosonde data is shown as red line. The blue bars represent the sampling fluctuations from 15:52 to 16:02 LST on 09 May, 2014.**

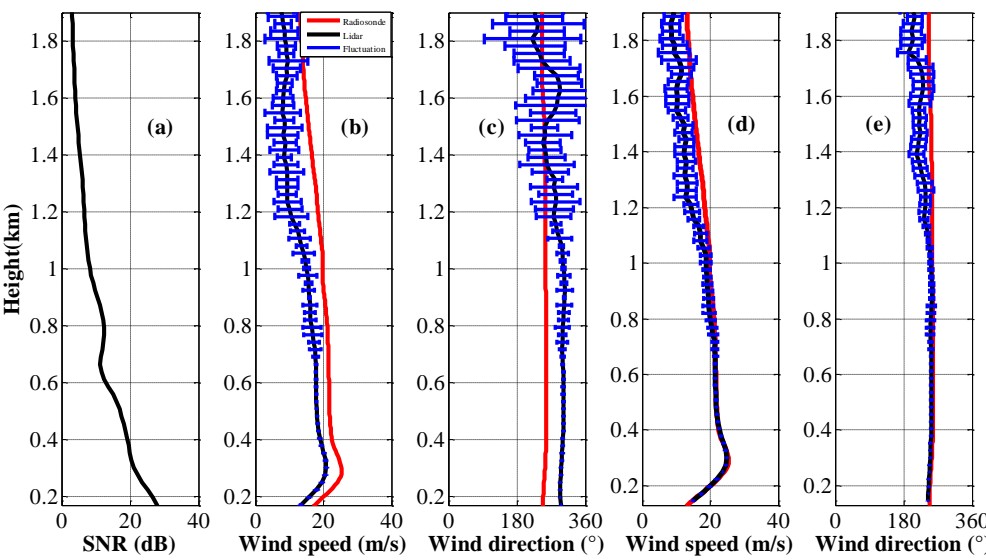

**Figure 6: As Fig. 5, but for 07:44 to 07:54 LST on 13 May, 2014 in cruising observation.**

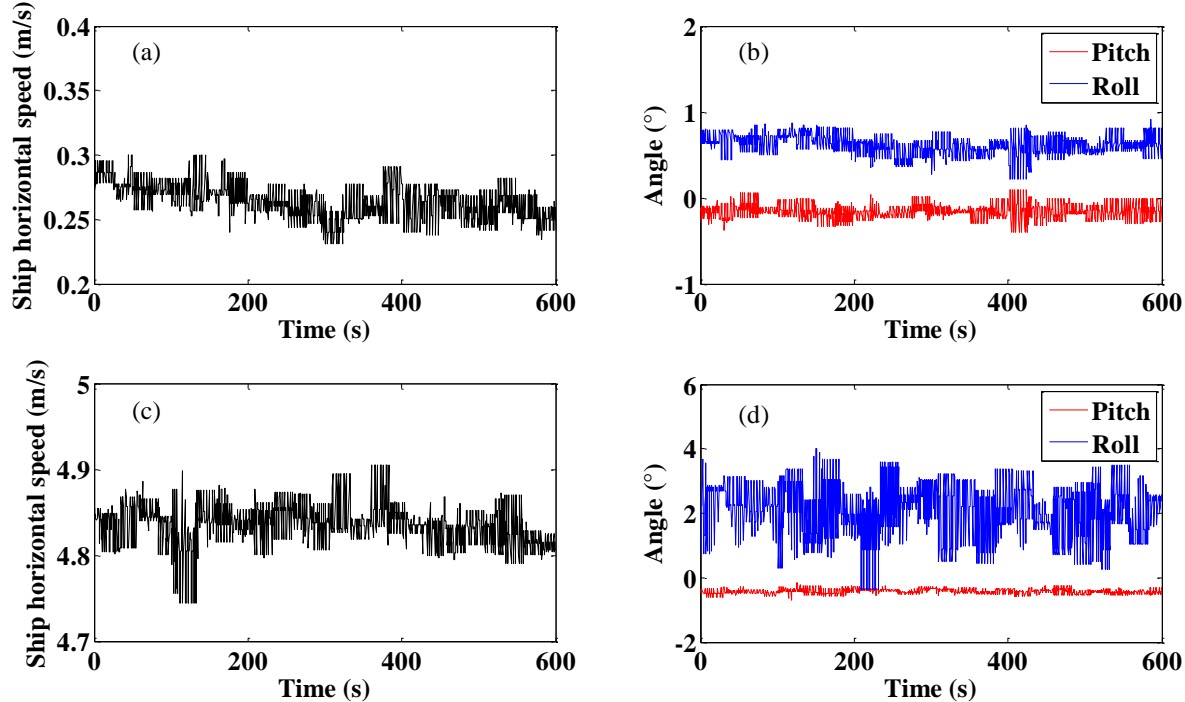

**Figure 7: (a) Time series of ship horizontal speed and (b) pitch and roll angles on 09 May 2014 (15:52-16:02) during anchored measurement, (c) Time series of ship horizontal speed and (d) pitch and roll angles on 13 May 2014 (07:44-07:54) during cruising measurement.**

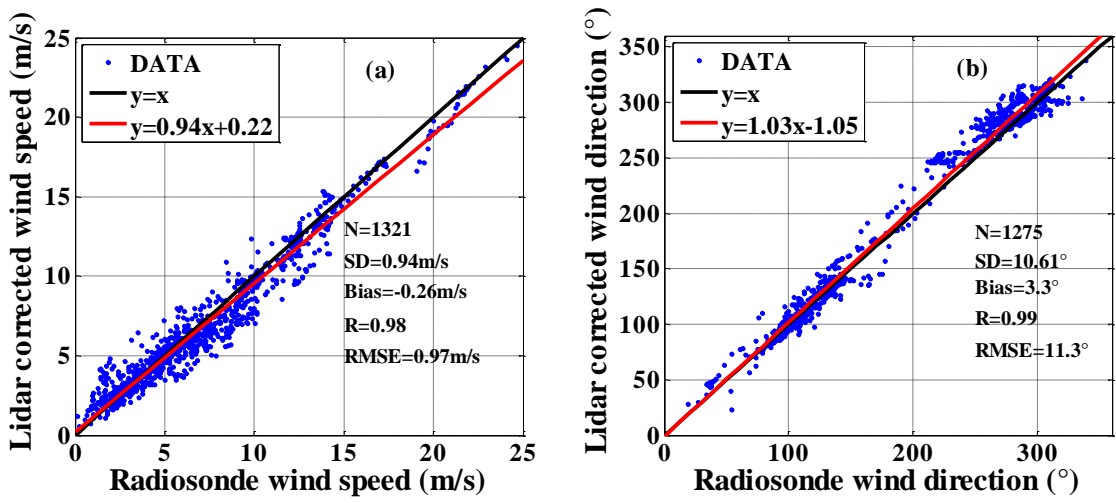

**Figure 8: Comparison of (a) wind speed and (b) wind direction between CDL and radiosonde data from 09 May 2014 to 19 May 2014. The number of points (N), standard deviation (SD), bias, correlation coefficient (R), and root-mean-square-error (RMSE) are also listed.**

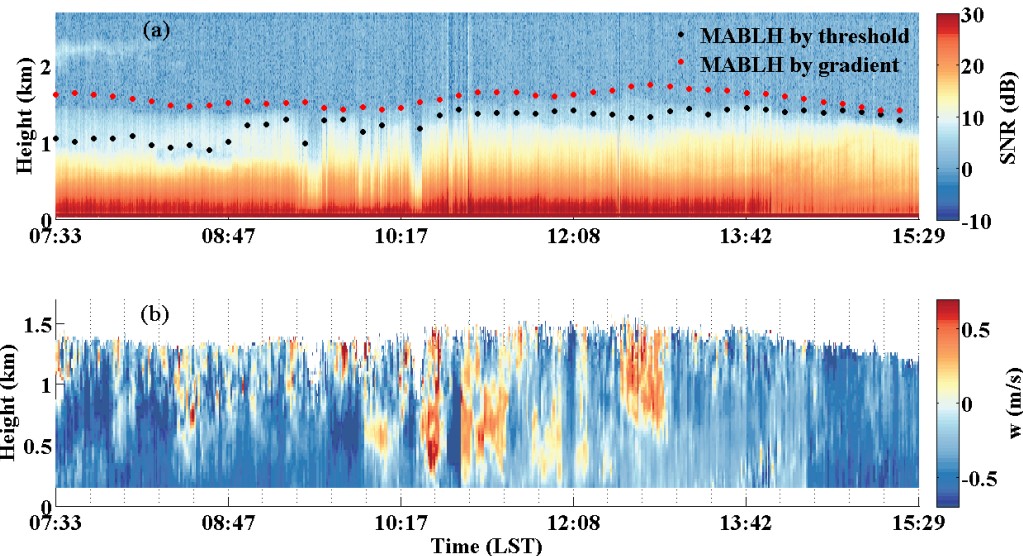

**Figure 9: Example measurement from 07:33 to 15:29 LST on 14 May 2014: (a) Time-Height-Intensity of SNR and retrieved MABL height using SNR threshold and gradient method (black and red solid circles, respectively). (b) Time-Height-Intensity of vertical velocity after attitude correction.**

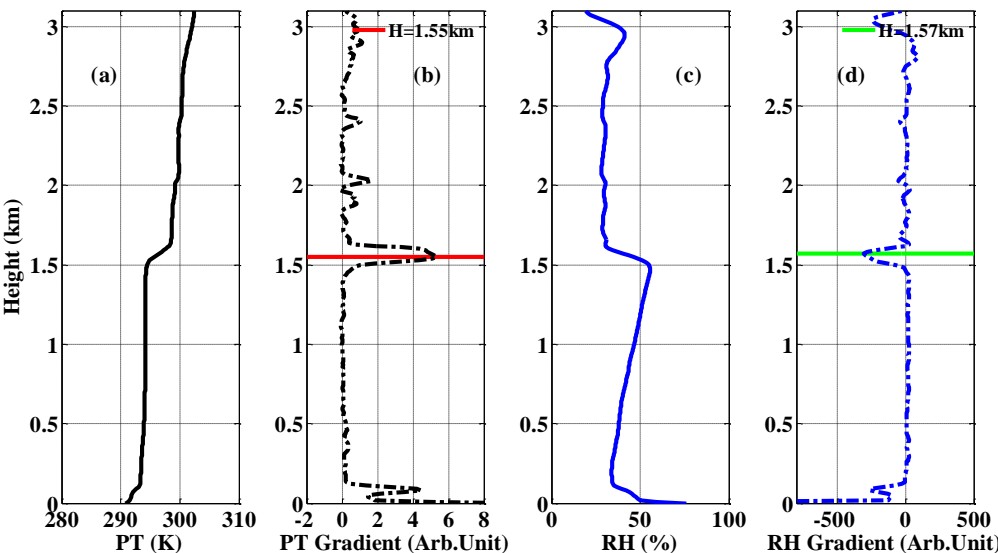

**Figure 10: Radiosonde profiles of (a) potential temperature (K) (b) the gradient of potential temperature (c) relative humidity (%) and (d) the gradient of relative humidity at 12:00, LST on 14 May 2014. The horizontal red and green lines in (b) and (c) stand for MABL height retrieved from potential temperature and relative humidity, respectively.**

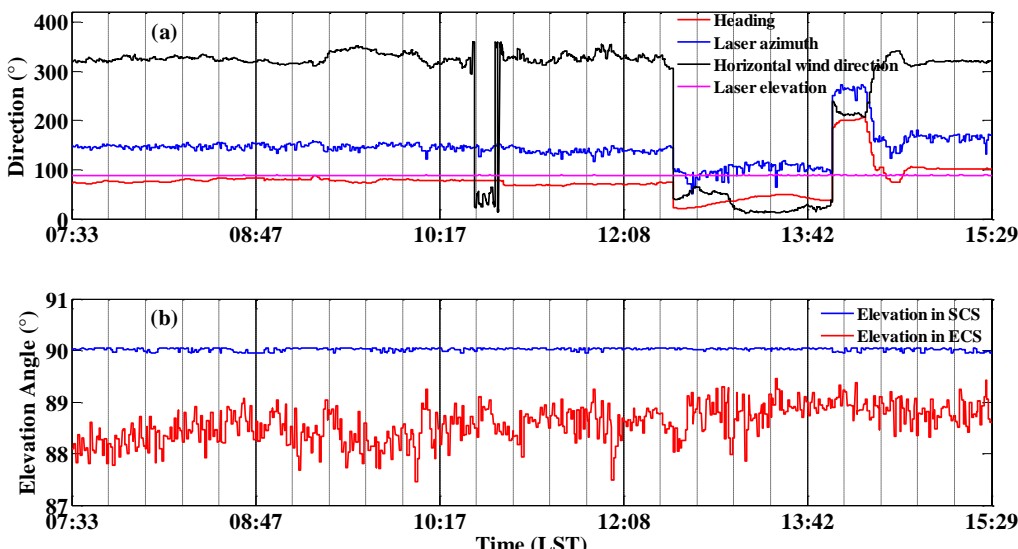

**Figure 11: Measurement from 07:33 to 15:29 LST on 14 May 2014: (a) Time series of ship heading, CDL laser beam azimuth and elevation in the Earth coordinate system, and horizontal wind direction at 0.4 km. (b) Elevation angle in zenith stare mode in Ship Coordinate System and Earth Coordinate System.**

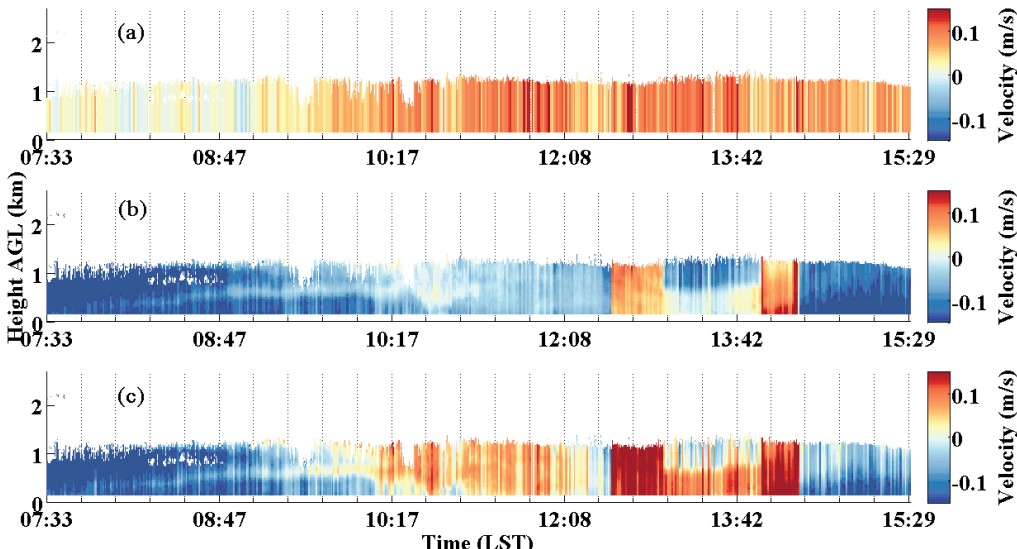

**Figure 12: Vertical velocity correction analysis: (a) projection of ship velocity on vertical velocity: $\vec{V}_{LOS\_ship}$ (b) the effect of horizontal wind on vertical velocity: $-\vec{r_g} \cdot \vec{V}$ (c) difference between vertical velocity after attitude correction and vertical velocity before attitude correction.**

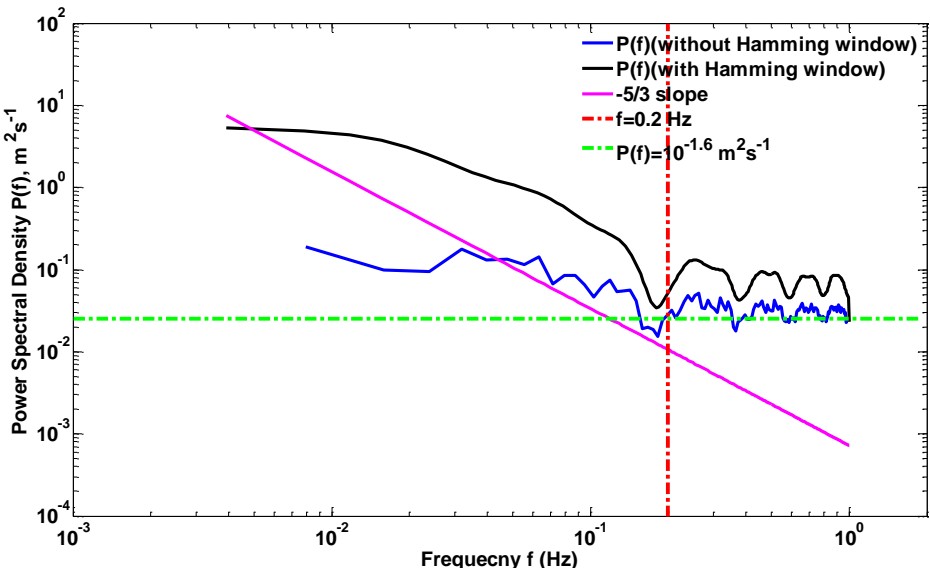

**Figure 13: Power spectrum density P(f) without and with Hamming window for the CDL measured vertical speed between 15:52 and 16:02 LST on 09 May and for an altitude of 1495 m (blue and black solid line, respectively). The expected spectrum behaviour according to the Kolmogorov's−5/3 law (pink solid line), the noise frequency threshold (red dotted line) and the derived noise floor for the CDL ( green dotted line) are shown.**

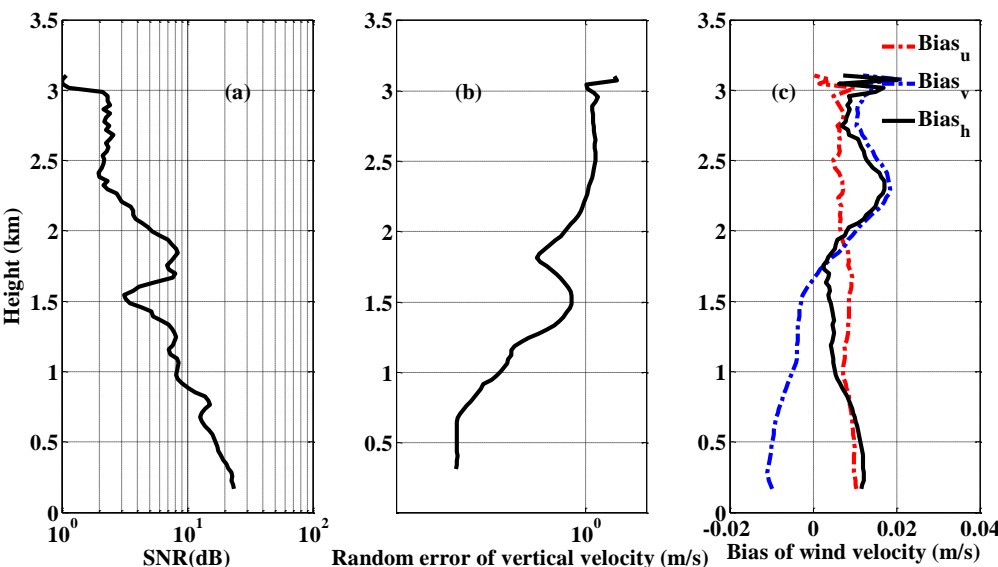

**Figure 14: The averaged profile of (a) SNR (b) Random error of vertical velocity (c) bias of horizontal wind north-south component (u), east-west component (v) and horizontal wind velocity (h) measured by CDL from 15:52 to 16:02 LST on 09 May, 2014.**

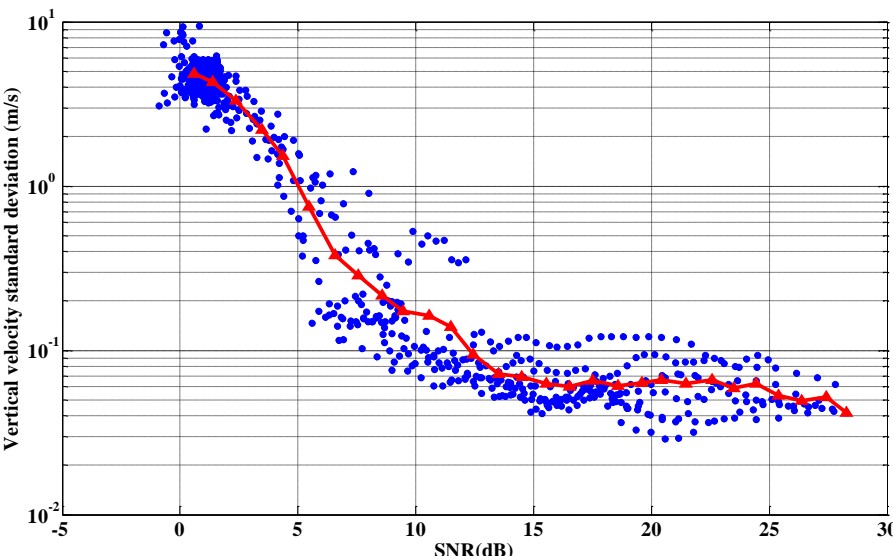

**Figure 15: Random error of the CDL vertical velocity from 07:33 to 15:29 LST on 14 May 2014 in all height range, which is determined as from the frequency spectrum of the retrieved vertical velocity. The averaged random error per SNR bin is shown in red-tringle line.**

**Table 1: Component Parameters of the CDL system**

| Qualification | Specification |
| --- | --- |
| Wavelength | 1.55 $\mu$m |
| Pulse repetition rate | 10 kHz |
| Pulse width | 100 ns - 400 ns |
| Pulse energy | 150 $\mu$J |
| Measurement range | 80 m - 4000 m (6000 m maximum) |
| Range resolution | 15 m - 60 m |
| Speed measurement uncertainty | $\leq 0.1$ ms$^{-1}$ |
| Radial velocity measurement range | $\pm37.5$ ms$^{-1}$ |
| Power Consumption | <300 W |
| weight | ~75 kg |
| Telescope diameter | 3 inches |
| Beam effective diameter | 60 mm |
| Focal length | 290 mm |

**Table 2: Component Parameters of the XW-GI5651 MEMS Inertial/Satellite Integrated Navigation system.**

| System real-time precision | | | | | |
|---|---|---|---|---|---|
| Heading | $0.1°$ (double antenna mode, baseline length $\geq 2$ m) | | | | |
| | $0.1°$ (single antenna, speed $> 10$ ms$^{-1}$) | | | | |
| Attitude | $0.1°$ | | | | |
| Position | Single point positioning $\leq 5$ m | | | | |
| | Real Time Kinematic (RTK) 2 cm + 1 ppm (Circular Error Probable, CEP) | | | | |
| Data updating rate | 200 Hz (configurable) | | | | |
| Starting time | $\leq 10$ s | | | | |
| Alignment time | 1~2 min (depending on dynamic maneuvering mode) | | | | |
| | Double antenna aided orientation time $\leq 1$ min | | | | |
| **Post-processing precision** | | | | | |
| Heading | $0.05°$ | | | | |
| Attitude | $0.05°$ | | | | |
| Position precision | Time to lose lock | 0 s | 10 s | 60 s | 300 s | 600 s |
| | Position | 0.02 m | 0.04 m | 3 m | 20 m | 60 m |
| **Physical properties** | | | | | |
| Power consumption | $< 7$ W | | | | |
| Working temperature | -40 °C ~ 80 °C | | | | |
| Overall size | 100 mm $\times$ 90 mm $\times$ 50 mm | | | | |
| Weight | $< 500$ g | | | | |

**Table 3: Component parameters of the GTS1 radiosonde**

| Meteorological Sensor | Specification | Technical Parameter |
|---|---|---|
| Temperature | Range | -90 - 50 °C |
| | Accuracy (standard deviation) | 0.2 °C (-80 - 50 °C) |
| | | 0.3 °C (-90 - -80 °C) |
| | Resolution | 0.1 °C |
| Humidity | Range | 0% RH - 100% RH |
| | Accuracy (standard deviation) | 5% RH ($T \geq 25\,^{\circ}C$) |
| | | 10% RH ($T \leq 25\,^{\circ}C$) |
| | Resolution | 1% RH |
| Pressure | Range | 1060 hPa - 5 hPa |
| | Accuracy (standard deviation) | 2 hPa (1050 hPa - 500 hPa) |
| | | 1 hPa (500 hPa - 5 hPa) |
| | Resolution | 0.1 hPa |

**Table 4: Ship motion parameters during anchored (first line) and cruising (second line) observations, respectively.**

| Date period | pitch | Roll | heading | Ship speed |
|---|---|---|---|---|
| 2014.05.09 15:52-16:02 | $-0.17^{\circ} \pm 0.06^{\circ}$ | $0.63^{\circ} \pm 0.11^{\circ}$ | $5.28^{\circ} \pm 1.22^{\circ}$ | $0.27\mathrm{ms}^{-1} \pm 0.01\mathrm{ms}^{-1}$ |
| 2014.05.13 07:44-07:54 | $-0.43^{\circ} \pm 0.05^{\circ}$ | $2.06^{\circ} \pm 0.87^{\circ}$ | $75.86^{\circ} \pm 1.22^{\circ}$ | $4.84\mathrm{ms}^{-1} \pm 0.03\mathrm{ms}^{-1}$ |

**Table 5: Statistics of the comparison between CDL and radiosonde at heights of 0.2, 0.4, 0.8, 1.2 and 1.6 km. Normalized RMSE is defined as RMSE divided by the maximum range of the measured values (maximum-minimum).**

| | Wind speed | | | | | Wind direction | | | | |
|---|---|---|---|---|---|---|---|---|---|---|
| Height (km) | 0.2 | 0.4 | 0.8 | 1.2 | 1.6 | 0.2 | 0.4 | 0.8 | 1.2 | 1.6 |
| Number points | 84 | 104 | 104 | 87 | 65 | 89 | 93 | 96 | 90 | 88 |
| SD ($ms^{-1}$)/($^\circ$) | 0.83 | 0.49 | 0.46 | 0.67 | 0.77 | 9.77 | 6.71 | 8.23 | 9.39 | 10.8 |
| Bias ($ms^{-1}$)/($^\circ$) | 0 | -0.1 | -0.3 | 0.26 | -0.5 | -3.4 | -2.7 | 0 | -0.1 | -6.3 |
| R | 0.97 | 0.99 | 0.99 | 0.98 | 0.98 | 0.99 | 0.99 | 0.99 | 0.99 | 0.98 |
| RMSE ($ms^{-1}$)/($^\circ$) | 0.83 | 0.50 | 0.59 | 0.72 | 0.94 | 10.3 | 7.22 | 8.18 | 9.34 | 12.5 |
| Normalized RMSE (%) | 4.6 | 2.3 | 3.3 | 5.9 | 7.4 | 4.3 | 3.2 | 3.2 | 3.4 | 6.8 |
| Slope | 1 | 0.99 | 1.04 | 1.09 | 1.10 | 0.99 | 1.01 | 1.01 | 1.09 | 1.10 |
| Intercept ($ms^{-1}$)/($^\circ$) | 0 | 0.2 | 0.01 | -0.9 | 0.03 | 4.23 | 0.77 | -1.7 | -4.9 | 5.1 |