# Peer review of "Shipborne Wind Measurement and Motion-induced Error Correction of a Coherent Doppler Lidar over the Yellow Sea in 2014"

_Atmospheric Measurement Techniques, 2017_

## Referee Comment (RC1) · Anonymous Referee #1 · 13 Oct 2017

General comments: This manuscript presents the results of wind measurements by coherent Doppler lidar from a ship in the Yellow Sea. The authors give a description of the algorithm for processing lidar data, which makes it possible to compensate for the measurement error associated with the motion of the ship. The results of joint measurements of height wind profiles by lidar and radiosonde are analyzed. The paper may be of interest to the readers of AMT. However, when describing the experiment and the data processing procedure, excessive attention is paid to secondary issues, and important details are ignored. Sometimes the terminology used by the authors makes it difficult to understand what they mean and how they obtained results presented in the manuscript. Some results raise doubts about their correctness. 1) The authors

assume that the bias of lidar estimate of the wind velocity is associated only with errors in determination of the ship speed and direction and with the pointing angle knowledge errors (Section 3.3). One can agree with this, if lidar estimates of the radial velocity are obtained at a sufficiently high signal-to-noise ratio (SNR, ratio of the signal spectrum peak to the standard deviation of noise component of the spectrum estimate), when the probability (or fraction) of a bad (unreliable) estimate of the radial velocity is practically zero. However, results shown in Fig.9 for heights above 2 km were obtained at SNR = 2 dB when the probability of bad estimate b = 0.3. As shown in Fig.4, true wind speed V = 5 m/s at a height of 2 km. According to the theory (Frehlich, R.G. and Yadlowsky, M.J.: Performance of mean-frequency estimators for Doppler radar and lidar, Journal of Atmospheric and Oceanic Technology 11(5), 1217-1230, 1994), the bias of velocity estimate BIAS = <Vˆ> - V, where <...> is ensemble averaging and Vˆ is the velocity estimate, is determined by the equation: BIAS = -b*V. Therefore at b = 0.3 and V = 5 m/s the bias equals -1.5 m/s. Nevertheless, in Fig. 9(ÑĄ) we see that the bias is about zero at SNR = 2. 2) Fig.9(b) shows the random error of wind velocity. On the other hand, the second term on the right-hand side of Eq.(11) is defined as the random error with zero mean. It is unclear how the result shown in Fig.9(b) was obtained. It is necessary to describe in more detail the procedure for obtaining this result. The results shown in Fig. 4 and Fig. 9 are obtained from the same lidar data (measurements from 15:52 to 16:02 on May 9, 2014)? 3) How can SNR be determined below 2 dB, if in this case with a high degree of probability the peak in the measured spectrum is associated with the noise, but not with the signal?

Specific Comments: 1) Page 3, lines 27-30: The pulse energy depends on the pulse width? If so, what is the pulse energy (and pulse repetition rate) for pulse durations of 100, 200 and 400 ns? 2) In section 2 the following information should be added: a) width of the time window (T) for obtaining the lidar signal power spectrum (T equals probing pulse duration of 200 ns?); b) width of the frequency band (B) within which the radial velocity was estimated from the lidar signal power spectrum (B = 50 MHz?); c) number of laser shots used for the spectral accumulation; d) number of radial velocity

estimates (for each range) that were obtained from lidar measurement for 10 minutes and then they were used for obtaining one estimate the wind vector. 3) Add the telescope diameter and beam diameter (1/e**2) to Table 1. 4) Page 8, lines 25-28: "It can been seen that the discrepancies in wind profile above 1 km between the radiosonde and lidar measurement are significant due to the multipath effect at the ship platform and decrease in collocation of the measurement." Another reason for the discrepancy between the results of the measurement of the wind by the lidar and the radiosonde at heights above 1 km is quite possible: the bias of the corrected lidar estimate of the wind due to the low SNR. It would be nice to add high profiles of the SNR in Figures 4 and 5. By the way, using some known procedure of filtration of good (reliable) estimates of the radial velocity obtained from 10-min lidar (4-DBS) measurements, the authors could obtain an unbiased wind speed estimate even in the case when the SNR is about 0 dB (if the percentage of good estimates is not below 20% ).

---

## Referee Comment (RC2) · Anonymous Referee #2 · 16 Oct 2017

Summary:

A newly developed ship-borne wind lidar, consisting of a coherent wind lidar from a Chinese manufacturer is presented in this manuscript. There are a few other papers on ship-borne wind lidars (e.g. Achtert et al. 2015 and from NOAA, e.g. Tucker et al. 2009) and thus this kind of application with its specific challenges (ship movement and environment) is challenging and still provides some novelty. In contrast to other earlier reports (e.g. Achtert et al. 2015), no active stabilisation of the complete lidar is performed, but the movement and angles are measured and corrected in post-processing. Some comparisons to radiosondes from a cruise in the Yellow Sea are shown in addition to two cases with vertical and horizontal wind measurements. Thus the topic of the manuscript fits to AMT. Major comments from my side are related to the description of the motion correction approach with GPS/INS, which is not clear at some places and lacks details to assess its novelty. Indeed the manuscript is very similar to the one of Achtert et al. (2015) in terms of description of methodology (correction algorithms), statistical comparison and evaluation with radiosonde, assessment of errors (spectral approach). Also numerous minor comments are related to the presentation of the topic. Thus I would recommend that the manuscript can be only accepted after major revisions of text, figures and additional material is included.

General and Major Comments:

1) The differences to the NOAA HRDL and the system by Achtert et al. 2015 should be mentioned more explicitly in the introductory paragraph (p. 3, 1st paragraph "it can be seen .." is not clear) Achtert et al. (2015) use an active motion-stabilized platform; so the difference to the described system here is clear. The NOAA HRDL uses a SDS to point the scanner LOS direction. But all systems need a motion-correction in the post-processing afterwards due to the limited accuracy of the active systems. So it is understood that the described system in the paper is neiter on a motion-stabilized platform nor the scanner LOS pointing direction is controlled by use of the ship attitude angles. Is this correct? If yes, then also the limitations of this approach (e.g. high ship movements, rough sea) should be discussed in the main part and summary more explicitly. On the other hand it is mentioned on p. 9, ch. 3.2 that, "the hemispherical scanner maintains the pointing of the lidar beam to zenith stare mode..". Does that mean that the scanner direction is controlled by the information from the INS?

2) The main part of the manuscript deals with the motion correction. Thus relevant parameters of the used GPS and INS system (type, accuracy, precision, data acquisition rate) should be provided and discussed. Why are 2 antennas shown in Fig. 1? Also the limitations of this approach, e.g. for high wind speeds or high angular rates during rough see conditions need to be discussed in the main text. Why did the authors not

chose an approach the control the scanner LOS direction, especially for the vertical pointing mode, by using the attitude angles from the INS (or is this applied)? Also details of the hard-target calibration need to be discussed. Is this performed once (before the cruise)? What angular offsets are determined, are different hard-targets in different direction used (range, elevation)? It is stated that "It can be seen that there exists no laser direction error ..". How do you come to this conclusion? Can you provide more details on that (e.g. data, Figure)?

3) The temporal resolution of the determination of the ship-induced Doppler shift (eq. 6) and the correction of the LOS velocity (eq. 7) needs to be stated and discussed. A figure showing a time-series of raw-data from the sensors (angles, velocity) could illustrate this to provide an impression about the time scales of the ship movement during anchored and cruising measurements. Also the timing of the DBS is not clear: How long is 1 LOS obtained, how long for the vertical velocity, and how long is the averaging time for the horizontal wind? Especially for the vertical pointing measurements the variability of the off-zenith angle should be shown in a Fig. The vertical velocity determination does need a correction for the horizontal wind. What is the time separation between the horizontal and vertical wind measurement?

4) In Achtert et al. (2015) the influence of the distortion of the flow due to the ship is discussed and modelled. In this manuscript this issue is only mentioned in 1-2 sentences on p. 9. What was the geometry/height of the ship? What would be the maximum height for a flow distortion, taking some numbers and scaling from the approach of Achtert et al (2015)? The lidar and radiosonde data is shown only above 150 m for this manuscript, but you conclude from your statistical comparison that the height of 200 m might be still affected by the flow distortion. So some more discussions on the geometry/height of the ship and the expected flow distortion around is needed.

5) Ch. 3.3. Error analysis: The authors deal here with the derivation of systematic errors (bias) to the horizontal wind retrieval I am wondering, if the error sources from the knowledge of the ship velocity and the lidar pointing angle are really systematic

(over longer timescales) or random, and would add to the random error of the wind retrieval. A clear distinction needs to be made in the underlying assumption for the ship velocity and lidar pointing wrt systematic and random errors. Are the provided numbers for ship velocity and pointing only the systematic part? What would be the random error of these quantities?

6) The authors could consider moving some of the equations related to the correction algorithms (Ch. 2) and error analysis (Ch. 3.3) to an appendix. At least for these parts, which are well known (e.g. coordinate transformations, descriptions of angles, DBS technique). I would restrict the description in ch. 2 and 3.3 to the novel aspects of this work.

7) I am missing a description of the overall objective of the deployment in the Yellow Sea in 2014 in the introduction. Was this only for technical demonstration, or were further atmospheric-oceanic processes studied. I am also missing a discussion of the open questions for turbulent flux measurements or wind vector measurements over the sea, which would need a shipborne Doppler wind lidar. One should discuss some objectives for the development of a shipborne wind lidar in the introduction. Also it might be useful to provide a paragraph in the Summary about future plans and campaigns.

Specific Comments

p.1 Intro: A number of studies are referenced for turbulent fluxes over the sea surface (Axford, 1968, …). Could these studies be grouped by objective, technology or geographical region to be more specific. Otherwise this long list of references is not very informative.

p.3, line 15: "Few studies .. in this region". Are there any references for these studies?

p. 5, L 11: The different elevation angles are probably due to ship rotation and movement during the time period of measuring different LOS directions, which should be stated here. Thus it is important to mention the duration of the measurement of each

LOS direction, and the complete 4 beams, and the relevant movements of the ship during these periods. How is the expected elevation angle $\theta 0$ obtained?

p. 7, 1st paragraph: It should be stated, how the background noise signal is obtained, e.g. via the recorded signal after a sufficiently long laser travel time, or via a separate measurement w/o laser pulse emission. Do the authors see an advantage of their SNR definition over the one from Banakh et al. 2013?

p. 7, L24: It should be described how the wind fluctuations are determined. Is it the standard deviation of wind measurements of higher temporal resolution (resolution?) during the 10 min.? Why are bars shown only for part of the profile in Fi.g 4 and 5? Is it smaller than a specific value below 1.4 km in Fig. 4? Do the fluctuations represent instrument noise or atmospheric fluctuations? What could be the reason that there are higher fluctuations in the layer of 1.4-1.6 km in Fig. 4?

p. 7, L27: Same question related to the method to determine the STD for the angles. Determined from the variability during the 10 min using raw data with of temporal resolution of xx s?

p.7, L28: Which SNR threshold was used here?

p.8, L26, last sentence: What is a "multipath effect"? This should be clarified. Also the difference in radiosonde and lidar location should be stated quantitatively. What is the difference in mean wind speed and direction between radiosonde and lidar above 1 km? Can a lidar instrumental effect excluded to explain the difference? I am not convinced that it is only colocation.

p.8 and Fig. 6: I would propose to plot the radiosonde on the x-axis and the lidar on the y-axis and also perform the linear least square fit with these coordinates. I consider the radiosonde as more accurate and the usual linear LSF procedures assume that the x-parameter is without errors (minimization of vertical differences). I also consider the criteria of excluding data with 1*SD as too strict. Only gross outliers – deviating from

a Gaussian distribution – could be excluded. This would typically result in a criteria of >3*SD. It needs also to be stated, how many data-pairs were excluded from the statistical comparison in order to judge the numbers of gross outliers. Also the SD typically refers to the SD of the difference (lidar-radiosonde). I am wondering how the SD of the lidar data ydata was obtained here. It is clear that the statistical parameters for bias, SD, R, and RMSE need to be calculated without a rigorous excluding of the data (with 1*SD). This point needs to be revisited and clarified.

p.9, ch. 3.2 and Fig. 7: The dots for MABL height are shown for the first 1/3 of Fig. 7 in a region of SNR around 10, where no obvious gradients can be seen, whereas for the second 2/3 it is more in the region between 10 dB (light blue) and 0 dB (dark blue). Please check and comment. Is there a reference about the ABL height determination using the first negative gradient?

p.11, L5: I consider an error of only 0.1° for the ship heading as very small. Is this justified by the hard-target measurements?

p.11, L8: What quantity is derived in eq. (14) in comparison to eq (13); Both are called "bias" LOS_N but eq. (14) with a "N'". Text should clearly state, the difference. What eq. (13 or 14) is then used in the estimates for the bias (eq. 17 and 18)?

p.11, eq. 15/16: These eq. could be moved to ch. 2 after eq (10), because it deals with u, and v retrieval and not with error estimates as in ch. 3.3.

p.11, L13: Here it is stated, that the lidar pointing angles are very small (and assumed to be perfect), but on p.12, L2 it is stated the errors are dominated by ship velocity and lidar pointing errors. This is in contradiction.

p.12, L6ff: Here the method of obtaining the random error is described ("In this case, a . . ."). But no resulting spectrum is shown in Fig. 9. This needs to be added or reformulated.

P12: L12: Are you sure that it is an elevated aerosol layer and not a cloud, which
provides the high SNR around 1.5 km?

p.12, L23: speckle-induced phase noise is not discussed in Achtert et al. 2015. Another reference needs to be provided

p.13 Summary: The limitations of the approach in comparison to existing systems need to be mentioned in the summary. Also I am missing an outlook about future algorithm or hardware improvements or future deployment during ship cruises.

p.13, L14: The number for the bias and the STD from the statistical comparison of all radiosondes should be stated here.

Ref. Liu et al. 2010: More details should be provided for this reference, which is not really accessible, or the reference should be removed or replaced. Also Achtert et al. (2015) provide these transformations.

Fig. 1: An additional Figure should be shown of the ship to illustrate the location of the CDL on the ship and possible disturbances of the flow.

Fig. 1: The location of the INS on the CDL should be indicated in the Figure.

Fig. 2: The symbols used for the angles pitch, roll, yaw should be placed also in the Figures.

Fig. 7: the legend within Fig. 7b is too small

Editorial: A large number of editorial comments were directly added to the PDF-Version of the manuscript. In addition the term "et al" needs to be replaced by "et al.". The manuscript needs thorough proof-reading after revision.

Please also note the supplement to this comment:
https://www.atmos-meas-tech-discuss.net/amt-2017-206/amt-2017-206-RC2-supplement.pdf

[Figure]

**Supplement:**

[Figure]

[Figure]

[revised manuscript text omitted]
$^{-1}$ )/( ° ) | 0 | 0.2 | 0.01 | -0.9 | 0.03 | 4.23 | 0.77 | -1.7 | -4.9 | 5.1 |

---

## Referee Comment (RC3) · Anonymous Referee #3 · 23 Oct 2017

Comment on "Shipborne Wind Measurement and Motion-induced Error Correction by Coherent Doppler Lidar over Yellow Sea in 2014", by X. Zhai, S. Wu, B. Liu, X. Song, J. Yin.

This manuscript describes a study that is relevant to Atmospheric Measurement Techniques. The authors describe procedures and measurement performance of a coherent Doppler wind lidar (CDWL) from ship. The manuscript proposes an algorithm to compensate for error of wind measurement due to the motion of the ship and provides contributions for lidar communities using shipborne applications. The manuscript has some issues that need clarification. There are numerous specific comments. A major

revision of manuscript is needed before it can be accepted for publication.

General comments: (1) Although details of a CDWL WindPrint S4000 are not described in the previous papers, in my opinion, details of the CDWL are not described well. The authors use an AOM for the heterodyne detection and a FPGA for FFT analysis. But there is no information about the sampling frequency and points used for FFT. The information is related to range-resolution for the time-domain, frequency-resolution for the frequency-domain and observable wind speed range. In the manuscript, bandwidth of 50MHz is used for data processing. Do you use a FPGA operating at a sampling frequency of 100MHz? Is it correct? In the previous paper, a AOM of 80MHz is used for the heterodyne detection. Therefore, frequency range of 60-100 MHz at center of 80MHz is detection range to determine LOS wind speed. 20MHz corresponds to be LOS wind speed of 15.5 m/s. ±50 m/s is "speed measurement range" shown in the previous paper. It is not consistent each other. It is puzzled to me. I might be missing something. . .and if so, please describe technical details and other aspects of the CDWL for better understanding the manuscript. (2) Definition of SNR is given in the manuscript. SNR=0dB means the signal power equals to noise power (NEP). Is it correct? Do you mean that the minus values are bad estimates? Minus values of SNR are shown in Figures 7 and 9. Why? Please describe details and derivation of SNR and search procedure by adding explanation sentences and figure. (3) It is also necessary to describe here the statistical process of these measures. How did you calculate LOS wind speed error and bias? How many radiosondes did you launch? What is the vertical resolution of the radiosonde? How do you interpolate to the CDWL data to for compare with the radiosonde? Ex. vertical resolution of the CDWL is 60m, while the vertical resolution of the radiosonde is 30m. The tow data points measured by the radiosonde are used for comparison with the CDWL. Or, one data averaged using two data points are used for that. Are data in the altitude range between 150m and 4000m used for the statistical comparison show in the Figure 6. A question comes for the difference between number N of 990 shown in Figure 6 and number points (wind speed: 84, 106. . .65; wind direction: 89,93. . .88) shown in Table 4. Why are the numbers

used for the wind speeds and wind directions are different? Please add explanation related to spatial and temporal difference between the DBS and the radiosonde measurements. (4) How did you determine misalignment and angle between the ship and laser beam axes? Although explanation sentences using a hard target are descried in the manuscript, technical details and other aspects of this are not described well. More details should be provided.

Specific Comments: P. 2 line 22, "wind speed". Do you mean "wind vector"? P. 3 line 4, "the CDL". a CDL P. 3 line 4, "vertical wind". horizontal wind P. 3 line 11, "In order to ...the Yellow sea". What is main purpose of the experimental investigation? Scientific or Engineering? P.3 line 30, "200ns". Is the number for $150\mu J$ at 10 KHz? P.4 line 2, "a proper". Do you mean "high"? P.4 line 9, "Yellow Sea". "the Yellow Sea". There are the same expressions in the manuscript. Please check in the manuscript P.4 line 13-15, "The inertial navigation system is ...laser beam". How did you confirm to keep the constant relative angle? P.4 line 16-17, "Hard target calibration". How did you conduct out the hard target calibration? Please describe details about it and statistical results (bias and error) P.4 line 19-20, "laser direction". Do you mean "laser beam direction"? What do you mean "between the ship and..."? Please explain it and show the axes in Figure 2. P.5 line 3, "the transmitting laser path". Do you mean "laser beam direction"? P.5 line 3, "the transmitting laser". Do you mean "laser beam direction"? P.5 Eq. 3, (x, y, z) ->(xg, yg, zg) P.5 line 9, "Where" -> "where" P.5 line 10, "Eq. (2)-(3)" -> "Eqs. (2) and (3)" P.5 Eq. 4, ( y/x) ->(yg / xg) P.5 Eq. 5, z -> zg P.7 line 3-4, please describe the definition of SNR using a figure. Does the definition of SNR have minus value? P.7 line 16-17, please give observation time to get each LOS wind speed profile. P.7 line 24, How do you determine the wind measurement fluctuation? P.7 line 28, "SNR threshold". Please add the threshold value. P.8 line 26, "measurement". "measurements" P.8 line 27, "multipath". I do not understand it. Please add explanations. P.11 line 1, "shipborne-based". "ship-based" or "shipborne". P.11 line 9-10, "assuming that the wind field has a constant horizontal and vertical velocity". Is the assumption always reasonable? What is the spatial and temporal

scale of wind field when the assumption is reasonable. P.11 line 13, "the lidar pointing angle". Do you mean the laser beam direction? P.12 line 6-8, "In this case...2016).". Why do you have to use the Hamming window and zero-padding? Please clarify it. What is the difference between with and without Hamming window and zero-padding? P.12 line 16. "Eq.". "Eqs." P.12 line 21-23. "8dB." Why "8dB" ? How do you determine the number? Please add explanation sentences. SNR looks the same value at 1 km as at 1.5 km. But different random errors at the altitudes are shown in the Figure 9. Why? P.12 line 23-25. "At reduced values...0 dB." Please plot results until SNR be 0dB in the Figures 9(a)-9(c). It is important for readers to identify the measurement performance of your CDWL. SNR=0 means the signal power equals to noise power (NEP), which is undetectable a true signal. Random errors would be large. "4 m/s" seems to be small. P.13 line 23-25. "At reduced values...0 dB." Please plot results in the figure until SNR be 0dB. P.14 line 4. "Shipborne wind observation". This manuscript describes an algorithm to compensate for error of wind measurement due to the motion of the ship, not observation. Please modify explanation sentences to insist on the main purpose of the manuscript. P.14 line 12. "The correlation...respectively,". Please add details such as date, time, altitude range, and so on. P.14 line 19-21. "random error...radiosonde data,". Please add explanation sentence about date, time and altitude.

Reference: P.14 line 12. Please add pages. P.14 line 23. Please add pages: 4675-4692.

Figure and Tables P.19 Figure 4. Label and "(a)" , "(b)", "(c)" and , "(d)" are small. Please use larger fonts. P.19 Figure 5. Label and "(a)" , "(b)", "(c)" and , "(d)" are small. Please use larger fonts. P.20 Figures 6. "(a)" and "(b)" are small. Please use larger fonts. P.20 Figures 7. Label and "(a)" , "(b)", and "(c)" are small. Please add horizontal wind speed. P.20 Figures 7. Label and "(a)" , "(b)", and "(c)" are small. Please use larger fonts. P.21 Figures 8. "(a)" , "(b)", and "(c)" are small. Please use larger fonts. P.21 Figures 9. "(a)" , "(b)", and "(c)" are small. Please use larger fonts.

---

## Author Comment (AC1) · 20 Nov 2017

Dear referee,

Thank you for your review of our manuscript. We greatly appreciate the substantial amount of time and effort that you dedicated to this review process.

We have revised the manuscript according to your comments and the point-by-point response is attached as supplement. The revised manuscript version is also provided at a separate PDF file.

Thanks again. Kind regards.

[Figure]

Please also note the supplement to this comment:
https://www.atmos-meas-tech-discuss.net/amt-2017-206/amt-2017-206-AC1-
supplement.zip

---

## Author Response (AR1)

Dear Editor and Referees,

Thank you for your review of our manuscript. We greatly appreciate the substantial amount of time and effort that you dedicated to this review process.

We have revised the manuscript according to your comments and the point-by-point response is also attached in this file. The marked-up manuscript version showing the changes made is also provided as follow. Please note that the numbers of tables and figures mentioned in this revised manuscript are different from those in the former manuscript.

In order to facilitate access to the response, the corresponding page number ranges are listed below and you can also link to the corresponding section by clicking the hyperlink:

1. Response to reviewer 1: Page 1 - Page 8;

2. Response to reviewer 2: Page 9 - Page 30;

3. Response to reviewer 3: Page 30 – Page 43;

4. Marked-up manuscript version: Page 44 – Page 85;

Thanks again.

Kind regards.

Songhua Wu

------------------------------------------Reviewer Comments----------------------------------------------

**1. Reviewer 1**

**General comments:**

This manuscript presents the results of wind measurements by coherent Doppler lidar from a ship in the Yellow Sea. The authors give a description of the algorithm for processing lidar data, which makes it possible to compensate for the measurement error associated with the motion of the ship. The results of joint measurements of height wind profiles by lidar and radiosonde are analyzed. The paper may be of interest to the readers of AMT. However, when describing the experiment and the data processing procedure, excessive attention is paid to secondary issues, and important details are ignored. Sometimes

the terminology used by the authors makes it difficult to understand what they mean and how they obtained results presented in the manuscript. Some results raise doubts about their correctness:

**1)** The authors assume that the bias of lidar estimate of the wind velocity is associated only with errors in determination of the ship speed and direction and with the pointing angle knowledge errors (Section 3.3). One can agree with this, if lidar estimates of the radial velocity are obtained at a sufficiently high signal-to-noise ratio (SNR, ratio of the signal spectrum peak to the standard deviation of noise component of the spectrum estimate), when the probability (or fraction) of a bad (unreliable) estimate of the radial velocity is practically zero. However, results shown in Fig.9 for heights above 2 km were obtained at SNR =2 dB when the probability of bad estimate b = 0.3. As shown in Fig.4, true wind speed V = 5 m/s at a height of 2 km. According to the theory (Frehlich, R.G. and Yadlowsky, M.J.: Performance of mean-frequency estimators for Doppler radar and lidar, Journal of Atmospheric and Oceanic Technology 11(5), 1217-1230, 1994), the bias of velocity estimate BIAS = <V^> - V, where <...> is ensemble averaging and V^ is the velocity estimate, is determined by the equation: BIAS = -b*V. Therefore at b = 0.3 and V = 5 m/s the bias equals -1.5 m/s. Nevertheless, in Fig. 9(Ñ, A) we see that the bias is about zero at SNR = 2.

R: I did not explain it clearly. The random error of radial velocity $e_V$ in an individual Doppler Lidar velocity estimate is dependent on the signal-to-noise ratio (SNR) of the measurement. It can be evaluated based on the frequency spectrum of the retrieved velocity, which is applied for vertical velocity random error estimation in this paper, and we can also use the velocity differences from even- and odd-numbered pulses to estimate the random error (Frehlich 2001). As for the random error of horizontal velocity, we need firstly obtain the random error of each radial velocity that is used for 4-DBS wind profile. Then the radial velocity error should be scaled into the horizontal velocity error based on the relationship between horizontal velocity components U, V and the radial velocity.

As for the bias in this paper, we deal with the derivation of systematic errors (bias) to the horizontal wind retrieval. The error sources from the knowledge of the ship velocity and the lidar pointing angle are systematic part, and it is assumed that the random error part of the ship velocity and the lidar pointing angle is zero, which is reasonable and robust for horizontal wind retrieval according to the specific parameters of lidar, GNSS and INS. The small bias at SNR=2 in Fig.9 actually represents the bias from the contribution of knowledge error of ship velocity and lidar pointing angle.

However, the bias of velocity estimates BIAS= <V^> - V, as the referee mentioned, is different from the definition in this paper. The definition of bias in this paper is $bias_v = \hat{V} - V_{truth} - e_V$, where $\hat{V}$ is the measured velocity estimate, $V_{truth}$ is the desired or true wind measurement and $e_V$ is the random error. It can be seen that the BIAS referee mentioned is the sum of

$bias_v$ and $e_v$ . As for the BIAS = -b*V=-0.3*5=-1.5 m/s, the referee mentioned, actually I really wonder how the b=0.3 is determined. It is mentioned in Frehlich's paper that the empirical model for the fraction of bad estimates b as a function of $\Phi$ for fixed $\Omega$ and M is: $b(\Phi) = [1+(\dfrac{\Phi}{b_0})^\alpha]^{-\gamma}$ . It is noted that the in this paper SNR is defined as the ratio of the peak value of FFT spectral signal in each range bin to the Root-Mean-Square (RMS) of background noise signal, which is different from Frehlich's paper definition and need to be treated carefully when determining those parameters. It would be possible to compare bias of two systems if we know the details of Frehlich's empirical model.

**2)** Fig.9 (b) shows the random error of wind velocity. On the other hand, the second term on the right-hand side of Eq. (11) is defined as the random error with zero mean. It is unclear how the result shown in Fig.9 (b) was obtained. It is necessary to describe in more detail the procedure for obtaining this result. The results shown in Fig. 4 and Fig. 9 are obtained from the same lidar data (measurements from 15:52 to 16:02 on May 9, 2014)?

R: Various methods of estimating the magnitude of the random error of Doppler Lidar velocity measurements have been introduced (Frehlich 2001). The measurements of error from velocity spectrum are used in this paper. A 50 % window overlap factor, a Hamming window is used in order to reduce the leakage in the spectra. A zero-padding of the missing values were applied to each window for each spectrum calculation to improve the frequency resolution. The constant high-frequency region of velocity spectrum higher than 0.2 Hz, shown in Figure 1 below, represents uncorrelated random error contribution, which is departing from the Kolmogorov's -5/3 law. The random error of vertical wind velocity is estimated as the standard deviation of the measured signal after high-pass filter.

[Figure]

Figure 1: Power spectral density P(f) without and with Hamming window for the CDL measured vertical speed between 15:52 and 16:02 LST on 09 May and for an altitude of 1495 m (blue and black solid line, respectively). The expected spectral behavior according to the Kolmogorov's−5/3 law (pink solid line), the noise frequency threshold (red dotted line) and the derived noise floor for the CDL ( green dotted line) are shown.

The results shown in Fig. 4 and Fig. 9 are obtained from the same lidar data (measurements from 15:52 to 16:02 on May 9, 2014).

**3)** How can SNR be determined below 2 dB, if in this case with a high degree of probability the peak in the measured spectrum is associated with the noise, but not with the signal?

R: The SNR in this study is defined as the ratio of the peak value of FFT spectral signal in each range bin to the Root-Mean-Square (RMS) of background noise signal. Figure 1 shows the array of the spectral $S(l\Delta f; k\Delta R)$, where $l = 0,1,2,3,...,L-1$ is the spectral channel number and $L = 100$. In this case the frequency resolution $\Delta f \approx 0.98$ MHz and the corresponding velocity resolution is $\Delta V = 0.76$ ms$^{-1}$. The bandwidth $B_{100} = (L-1)\Delta f = 97.68$ MHz, and the corresponding radial velocity measurement range is $\pm 37.5$ ms$^{-1}$. Figure 1a shows the last 10 range gates raw array of spectral in green line. We estimate the averaged background noise spectrum

$$\overline{S}_N(l\Delta f) = \frac{1}{10}\sum_{k=94}^{103} S(l\Delta f; k\Delta R) \tag{8}$$

Subtracting the background noise spectral $\overline{S}_N(l\Delta f)$ from the raw spectral array $S(l\Delta f; k\Delta R)$, the unnoisy array of spectral $S(l\Delta f; k\Delta R)$ can be obtained and shown in red line in Fig. 1. The peak value index $l_{peak}$ from the $S(l\Delta f; k\Delta R)$ can be firstly obtained and thus the absolute signal power $P_s(k\Delta R)$ at various ranges $k\Delta R$ can be represented as:

$$P_s(k\Delta R) = S(l_{peak}\Delta f; k\Delta R) - \frac{1}{12}\left(\sum_{l_{peak}-20}^{l_{peak}-15} S(l\Delta f; k\Delta R) + \sum_{l_{peak}+15}^{l_{peak}+20} S(l\Delta f; k\Delta R)\right) \tag{9}$$

Replacing integration by summation and taking into account that the zero velocity point in one channel is $l_{zero} = 50$, we estimate the noise power $P_N$ as

$$P_N = \frac{1}{10}\sum_{k=94}^{k=103} \sqrt{\frac{1}{21}\sum_{l=l_{zero}-10}^{l_{zero}+10} \hat{S}_N(l\Delta f; k\Delta R)^2} \tag{10}$$

Finally, we obtain the range profile of the $SNR(k\Delta R)$ using the equation

$$SNR(k\Delta R) = 10\log_{10}(\frac{P_s(k\Delta R)}{P_N}) \tag{11}$$

[Figure]

Figure 1: The CDL measured array of the FFT spectra (a) the last 10 range gates spectra for background noise spectrum estimation (b) the 1st – 5th range gates (150 m – 270 m, range resolution is 30 m) spectrum.

5   The SNR from Banakh et al. 2013 is defined as the ratio of the averaged heterodyne signal power $P_s$ to the average detector noise power $P_n$ in a 50-MHz bandwidth. The power $P_s$ and $P_n$ are integrals of the spectral densities $S_s(f)$ and $S_n(f)$, respectively, in frequency f within a band of width $B_{50}$, that is:

$$P_s = \int_{B_{50}} S_s(f)df \tag{5}$$

$$P_n = \int_{B_{50}} S_n(f)df \tag{6}$$

Comparing the definition from Banakh et al. 2013, the SNR in this paper is simpler and also indicates the CDL detection
10   capability, data accuracy and atmospheric tracer particle relative intensity. In this sense, the SNR threshold value in this paper is higher than the one in previous studies (Banakh et al. 2013; Achtert et al 2015) for the same signal power spectrum.

**Specific comments**

**1)** Page 3, lines 27-30: The pulse energy depends on the pulse width? If so, what is the pulse energy (and pulse repetition rate)
15   for pulse durations of 100, 200 and 400 ns?

R: I did not explain it clearly. The pulse energy is fixed and the pulse width is configurable. Pulse width is the full width at half maximum of the laser pulse waveform. A wider pulse width results in a larger measurement blind spot, but increases the average power and detection distance of the laser. Narrower pulse width can reduce the measurement of blind spots, but must also reduce the average power of the laser in order to control the laser peak power within the maximum range of the fiber, that
20   is, will reduce the detection range.

The description in the revised manuscript is: "The achieved pulsed energy is approximately 150 μJ and the pulse repetition frequency is 10 kHz ."

**2)** In section 2 the following information should be added:

a) width of the time window (T) for obtaining the lidar signal power spectrum (T equals probing pulse duration of 200 ns?);

R: T=200 ns.

"The pulse width produced by the modulation, which is also the width of time window for obtaining the lidar signal power spectrum, is adjustable from 100 ns to 400 ns , thus the spatial resolution can be varied from 15 m to 60 m . We typically operate the CDL with a pulse width of 200 ns in this study."

b) width of the frequency band (B) within which the radial velocity was estimated from the lidar signal power spectrum (B = 50 MHz?);

R: The bandwidth $B_{100} = (L-1)\Delta f = 97.68$ MHz is used for radial velocity estimation. The specific introduction can be seen in the answer to general comment question 3.

c) number of laser shots used for the spectral accumulation;

R: N=5000

d) number of radial velocity estimates (for each range) that were obtained from lidar measurement for 10 minutes and then they were used for obtaining one estimate the wind vector.

R: Both the determination of the ship-induced Doppler shift and the radial velocity have the same temporal resolution of 0.5 s. Figure 4 in the revised manuscript shows the flowchart of shipborne CDL data processing. Specifically, the LOS velocity and Signal to Noise Ratio (SNR) can be firstly determined using lidar data and FFT analysis. After the data pre-processing including the quality control based on SNR threshold, the attitude transformation is then used to obtain the azimuth and elevation in each LOS vector in Earth coordinate system with temporal resolution of 0.5 s. The LOS velocity detected by lidar is the atmosphere motion relative to ship coordinate system, thus the removal of the along-beam platform velocity due to ship motion is needed. In this study, the horizontal wind profile with 2-min temporal resolution will be retrieved for vertical velocity correction. Basically, the LOS velocities from $N$, $S$, $E$, and $W$ direction after SNR quality control during the chosen 2-min interval are collected firstly. Then the procedure of filtration of reliable estimates of each radial velocity based on SNR threshold is used to obtain "good" speed estimates. The selected radial velocities and corresponding ship condition information in each radial direction are averaged and the averaged ship condition will be used for the removal of platform velocity effect. Finally, the horizontal with 2-min temporal resolution can be retrieved using modified 4-DBS mode. The vertical wind measurement has a temporal resolution of 0.5 s, the horizontal wind whose retrieved time is closest to vertical wind measured time will be used for vertical velocity correction.

**3)** Add the tele-scope diameter and beam diameter (1/e**2) to Table 1

R: The parameters have added to the Table 1 in revised version.

| Telescope diameter | 3 inches |
|---|---|
| Beam effective diameter | 60 mm |
| Focal length | 290 mm |

**4)** Page 8, lines 25-28: "It can been seen that the discrepancies in wind profile above 1 km between the radiosonde and lidar measurement are significant due to the multipath effect at the ship platform and decrease in collocation of the measurement."

Another reason for the discrepancy between the results of the measurement of the wind by the lidar and the radiosonde at heights above 1 km is quite possible: the bias of the corrected lidar estimate of the wind due to the low SNR. It would be nice to add high profiles of the SNR in Figures 4 and 5. By the way, using some known procedure of filtration of good (reliable) estimates of the radial velocity obtained from 10-min lidar (4-DBS) measurements, the authors could obtain an unbiased wind speed estimate even in the case when the SNR is about 0 dB (if the percentage of good estimates is not below 20% ).

R: It can be seen that the discrepancies in wind profile above 1 km between the radiosonde and lidar measurement are significant. On the one hand, the random error of the corrected CDL estimation of the wind due to the low SNR shown in Fig. 6a contributes to this discrepancy. On the other hand, the drift of radiosonde is affected by atmospheric turbulence perturbations and the CDL detection volume is changing during cruising observation. The spatial separation between radiosonde and CDL which can be called multipath effect, can cause larger discrepancy with increasing altitude.

Thanks for your suggestions, the SNR profile has added to the Fig 5 and 6 in the revised version, as shown below:

[Figure]

Figure 1: Anchored observation: (a) SNR profile (b) (c) wind speed and (d) (e) wind direction measured by CDL (blue line) before and after attitude correction, respectively. The simultaneous radiosonde data is shown in red line. The blue bars represent the sampling fluctuations from 15:52 to 16:02 LST, 09 May, 2014.

[Figure]

Figure 2: As Fig. 1, but for 07:44 to 07:54 LST 13 May, 2014 in cruising observation.

**2. Reviewer 2**

**Summary:**

A newly developed ship-borne wind lidar, consisting of a coherent wind lidar from a Chinese manufacturer is presented in this manuscript. There are a few other papers on ship-borne wind lidars (e.g. Achtert et al. 2015 and from NOAA, e.g. Tucker et al. 2009) and thus this kind of application with its specific challenges (ship movement and environment) is challenging and still provides some novelty. In contrast to other earlier reports (e.g. Achtert et al. 2015), no active stabilisation of the complete lidar is performed, but the movement and angles are measured and corrected in post-processing. Some comparisons to radiosondes from a cruise in the Yellow Sea are shown in addition to two cases with vertical and horizontal wind measurements. Thus the topic of the manuscript fits to AMT. Major comments from my side are related to the description of the motion correction approach with GPS/INS, which is not clear at some places and lacks details to assess its novelty. Indeed the manuscript is very similar to the one of Achtert et al. (2015) in terms of description of methodology (correction algorithms), statistical comparison and evaluation with radiosonde, assessment of errors (spectral approach). Also numerous minor comments are related to the presentation of the topic. Thus I would recommend that the manuscript can be only accepted after major revisions of text, figures and additional material is included.

**General and Major Comments:**

**1)** The differences to the NOAA HRDL and the system by Achtert et al. 2015 should be mentioned more explicitly in the introductory paragraph (p. 3, 1st paragraph "it can be seen .." is not clear) Achtert et al. (2015) use an active motion-stabilized platform; so the difference to the described system here is clear. The NOAA HRDL uses a SDS to point the scanner LOS direction. But all systems need a motion-correction in the post-processing afterwards due to the limited accuracy of the active systems. So it is understood that the described system in the paper is neither on a motion-stabilized platform nor the scanner LOS pointing direction is controlled by use of the ship attitude angles. Is this correct? If yes, then also the limitations of this approach (e.g. high ship movements, rough sea) should be discussed in the main part and summary more explicitly. On the other hand it is mentioned on p. 9, ch. 3.2 that, "the hemispherical scanner maintains the pointing of the lidar beam to zenith stare mode..". Does that mean that the scanner direction is controlled by the information from the INS?

R: The described system in the paper is neither on a motion-stabilized platform nor the scanner LOS pointing direction is controlled by use of the ship attitude angles. The LOS velocity measured by CDL in ship coordinate system $\vec{V}_{LOS\_measure}$ is unaffected by the ship movement, therefore the approach is available under high ship movement. $\vec{V}_{LOS\_measure}$. Since the bandwidth $B_{100} = (L-1)\Delta f = 97.68$ MHz, the corresponding radial velocity measurement range is $\pm 37.5$ ms$^{-1}$. As for the feasibility under different sea condition, generally, except for the extremely rough sea condition, the LOS velocity component from vertical velocity in different directions is assumed to be identical. Then the $u$, $v$ can be calculated using a modified 4-DBS formula. Under extremely rough sea condition, the difference of elevation angle in different directions is significant, and

the contribution of vertical velocity to LOS velocity needed to be treated carefully. In this case, the height interpolation of radial velocity can be used, and if three or more radial velocities at the same height are obtained, the horizontal and vertical velocity can be retrieved. But if the elevation angle in one direction is too small, the detectable height will be limited. Figure 1 shows the statistical distribution of the lidar pitch and roll angle from 09 May 2014 to 19 May 2014. In most cases the sea condition is less rough and the approach can be used reasonably.

[Figure]

Figure 1. Statistical distribution of the lidar pitch and roll angle from 09 May 2014 to 19 May 2014.

I didn't explain it clearly, "the hemispherical scanner maintains the pointing of the lidar beam to zenith stare mode", in this sentence, the "zenith stare mode" represents the measurement in Lidar coordinate system, not the scanner direction in ECS controlled by the information from the INS

**2)** The main part of the manuscript deals with the motion correction. Thus relevant parameters of the used GPS and INS system (type, accuracy, precision, data acquisition rate) should be provided and discussed. Why are 2 antennas shown in Fig. 1? Also the limitations of this approach, e.g. for high wind speeds or high angular rates during rough see conditions need to be discussed in the main text. Why did the authors not chose an approach the control the scanner LOS direction, especially for the vertical pointing mode, by using the attitude angles from the INS (or is this applied)? Also details of the hard-target calibration need to be discussed. Is this performed once (before the cruise)? What angular offsets are determined, are different hard-targets in different direction used (range, elevation)? It is stated that "It can be seen that there exists no laser direction error." How do you come to this conclusion? Can you provide more details on that (e.g. data, Figure)?

[revised manuscript text omitted]

**3)** The temporal resolution of the determination of the ship-induced Doppler shift (eq. 6) and the correction of the LOS velocity (eq. 7) needs to be stated and discussed. A figure showing a time-series of raw-data from the sensors (angles, velocity) could illustrate this to provide an impression about the time scales of the ship movement during anchored and cruising measurements. Also the timing of the DBS is not clear: How long is 1 LOS obtained, how long for the vertical velocity, and how long is the averaging time for the horizontal wind? Especially for the vertical pointing measurements the variability of the off-zenith angle should be shown in a Fig. The vertical velocity determination does need a correction for the horizontal wind. What is the time separation between the horizontal and vertical wind measurement?

Reply: Both the determination of the ship-induced Doppler shift and the radial velocity have the same temporal resolution of 0.5 s. Figure 4 in the revised version shows the flowchart of shipborne CDL data processing. Specifically, the LOS velocity and Signal to Noise Ratio (SNR) can be firstly determined using lidar data and FFT analysis. After the data pre-processing including the quality control based on SNR threshold, the attitude transformation is then used to obtain the azimuth and elevation in each LOS vector in Earth coordinate system with temporal resolution of 0.5 s. The LOS velocity detected by lidar is the atmosphere motion relative to ship coordinate system, thus the removal of the along-beam platform velocity due to ship motion is needed. In this study, the horizontal wind profile with 2-min temporal resolution will be retrieved for vertical velocity correction. Basically, the LOS velocities from $N$, $S$, $E$, and $W$ direction after SNR quality control during the chosen 2-min interval are collected firstly. Then the procedure of filtration of reliable estimates of each radial velocity based on SNR threshold is used to obtain "good" speed estimates. The selected radial velocities and corresponding ship condition information in each radial direction are averaged and the averaged ship condition will be used for the removal of platform

velocity effect. Finally, the horizontal with 2-min temporal resolution can be retrieved using modified 4-DBS mode. The vertical wind measurement has a temporal resolution of 0.5 s, the horizontal wind whose retrieved time is closest to vertical wind measured time will be used for vertical velocity correction. A time-series of raw-data from the sensors (angles, velocity) and corrected angles can be seen in Fig.1 and Fig 2 as below:

[Figure]

Figure 1: (a) Time series of ship horizontal speed and (b) pitch and roll angles on 09 May 2014 (15:52-16:02) during anchored measurement, (c) Time series of ship horizontal speed and (d) pitch and roll angles on 13 May 2014 (07:44-07:54) during cruising

[Figure]

measurement.

Figure 2: Example measurement from 07:33 to 15:29 LST 14 May 2014: (a) Time series of ship heading, CDL laser beam azimuth and elevation in the Earth coordinate system, and horizontal wind direction at 0.4 km. (b) Elevation angle in zenith stare mode in Ship Coordinate System and Earth Coordinate System.

5   **4)** In Achtert et al. (2015) the influence of the distortion of the flow due to the ship is discussed and modelled. In this manuscript this issue is only mentioned in 1-2 sentences on p. 9. What was the geometry/height of the ship? What would be the maximum height for a flow distortion, taking some numbers and scaling from the approach of Achtert et al (2015)? The lidar and radiosonde data is shown only above 150 m for this manuscript, but you conclude from your statistical comparison that the height of 200 m might be still affected by the flow distortion. So some more discussions on the geometry/height of the ship

10  and the expected flow distortion around is needed.

R: The height of Dongfanghong-2 is 84 m. The relative height between CDL and the highest building on ship is about 15 m shown in Figure 1. When the strong wind blows from the ship bow, the building and experimental setups on ship have an important effect on CDL lower-level detection volume where the induced-turbulence may cannot meet the assumption of

15  homogeneous isotropic atmosphere for 4-DBS retrieval. On the other hand, the blind area of CDL is 150 m and corresponds to the height of 129.9 m when laser beam elevation angle is $60^{\circ}$, meaning that less data points are available below 200 m with effective comparison. Therefore, whether the flow distortion around the ship is the main reason for the discrepancies in the lower part measurement or not is yet unclear. Further study, especially focused on the CFD model, needs to be used to the assess the potential effects on turbulent flow and wind field analysis.

20  In Achtert's paper, it is concluded that the normalized bias in horizontal wind speed is less than 2% for all wind directions at altitudes above 75m. But the specific geometric parameters of the ship in CFD simulation domain are not mentioned, which is important for determination of the maximum height for a flow distortion induced by ship. However, it surely provides us a new sight for Lidar data quality assessment, especially for the correction of the wind measurements used for turbulence fluxes exchange from Marine-Atmosphere interface.

[Figure]

Figure 1. The Dongfanghong-2 research vessel during 2014 Yellow Sea Campaign. The red solid dot represents the CDL position.

**5)** Ch. 3.3. Error analysis: The authors deal here with the derivation of systematic errors (bias) to the horizontal wind retrieval I am wondering, if the error sources from the knowledge of the ship velocity and the lidar pointing angle are really systematic (over longer timescales) or random, and would add to the random error of the wind retrieval. A clear distinction needs to be made in the underlying assumption for the ship velocity and lidar pointing wrt systematic and random errors. Are the provided numbers for ship velocity and pointing only the systematic part? What would be the random error of these quantities?

R: It is noted that the knowledge error of the ship velocity and lidar pointing angle mentioned in Part 3.3 are systematic part and it is assumed that the random error of these parameters is zero, which is reasonable and robust for horizontal wind retrieval.

**6)** The authors could consider moving some of the equations related to the correction algorithms (Ch. 2) and error analysis (Ch. 3.3) to an appendix. At least for these parts, which are well known (e.g. coordinate transformations, descriptions of angles, DBS technique). I would restrict the description in ch. 2 and 3.3 to the novel aspects of this work.

R: Thanks for your suggestion. The motion-correction algorithm including Eq. 1-5 in the manuscript has moved to Appendix A.

**7)** I am missing a description of the overall objective of the deployment in the Yellow Sea in 2014 in the introduction. Was this only for technical demonstration, or were further atmospheric-oceanic processes studied. I am also missing a discussion of the open questions for turbulent flux measurements or wind vector measurements over the sea, which would need a shipborne Doppler wind lidar. One should discuss some objectives for the development of a shipborne wind lidar in the introduction. Also it might be useful to provide a paragraph in the Summary about future plans and campaigns.

R: Thanks for your suggestion. The description of the objectives has been added to the last paragraph in the Introduction part. It is described below:

"The experimental investigation was undertaken by Dongfanghong-2 research vessel affiliated with Ocean University of China in 2014 over the Yellow Sea. The Yellow Sea, a marginal sea of the Pacific Ocean, is the northern part of the East China Sea. It is located between mainland China and the Korean Peninsula. There is seldom study on boundary layer dynamics study based on CDL in this region. As one of the main objectives, the CDL was deployed on the ship in this campaign to demonstrate the feasibility of the algorithm-based attitude correction method. The obtained accurate three-dimensional wind information can provide significant preparation for further studies on characteristics of dynamics and thermodynamics in the MABL and turbulence flux exchange over sea surface. In addition to CDL, as another important part of this campaign, a High Spectral Resolution Lidar (HSRL) and a CL31 ceilometer were also deployed on the ship platform in order to detect MABL height spatial-temporal evolution and to retrieve the aerosol and cloud optical characteristics such as extinction coefficient and backscatter ratio and so forth. It will help us to understand the complex behavior of MABL and the aerosol cloud forcing

characteristics over sea region and the impact on climate change. This paper focuses on CDL performance and gives a thorough analysis of the attitude correction for lidar velocity measurement."

The description of the further plan has been added to the last paragraph in the Summary part. It is described below:

"Overall, combining a CDL with attitude correction system and accurate motion correction process as presented here forms a reliable and autonomous set-up that could be placed on mobile platform to provide more detailed, higher spatial and temporal resolution view of three-dimensional wind field information. It will be further validated and improved under different sea conditions using CFD model simulation and field campaign. More specific studies are being carried out or prepared, including

10 atmospheric turbulence characteristics statistics and multi-scale wind field observation in MABL, wind turbine wake and atmospheric turbulence interaction over offshore wind power field (Wu et al., 2016; Zhai et al., 2017), mass transport and flux analysis in MABL with combination of CDL and Multi-wavelength Polarization Raman Lidar (Wu et al., 2016)."

**Specific Comments**

**1)** p.1 Intro: A number of studies are referenced for turbulent fluxes over the sea surface (Axford, 1968). Could these studies be grouped by objective, technology or geographical region to be more specific. Otherwise this long list of references is not very informative.

R: Thanks for your suggestion. The references has been grounded according to different platforms and geographical region. It

20 is shown below:

"There are many studies on the turbulent fluxes measurement over the sea surface. Various motion sensing technique on the moving platform has been developed in the field of airborne (Axford, 1968), space-borne (Hawley et al., 1993) and shipborne observations (Fujitani, 1992; Song et al., 1996; Edson et al., 1998; Miller et al., 2008). Many shipborne field experiments have

25 been widely carried out over Pacific Oceanic area (Mitsuta et al., 1974; Bradley et al., 1991; Shao, 1995; Tsukamoto et al., 1995)."

**2)** p.3, line 15: "Few studies .. in this region". Are there any references for these studies?

R: As far as we know, there is seldom study on boundary layer dynamics study based on CDL in this region.

**3)** p. 5, L 11: The different elevation angles are probably due to ship rotation and movement during the time period of measuring different LOS directions, which should be stated here. Thus it is important to mention the duration of the measurement of each

LOS direction, and the complete 4 beams, and the relevant movements of the ship during these periods. How is the expected elevation angle _0 obtained?

R: Thanks for your suggestion, the description has been added in the manuscript. It is shown below:

5    But for the shipborne platform, the elevation $\theta_g$ in four directions (north, south, west and east in ship coordination system) may have slightly difference (see Eq. (A5)) due to ship rotation and movement during the time period of measuring different LOS directions, thus a conversion of $\vec{V}_{LOS}$ from real elevation $\theta_g$ to the expected elevation $\theta_0$ is firstly processed, that is,

$$\vec{V}_{LOS}' = \vec{V}_{LOS} \cos\theta_0 / \cos\theta_g \qquad (4)$$

It is noted that $\theta_0$ can be set any value from $0°$ to $90°$, and in this paper $\theta_0 = 60°$ is set for horizontal wind profile retrieval. During the experiment, each radial direction will take 5 s to obtain 10 measured LOS velocity for accumulation and average.
10   In this sense, the highest temporal resolution of horizontal wind velocity using 4-DBS mode is 20 s. The recorded ship condition information has the same update rate of 0.5 s as radial velocity's, which can be averaged to remove the platform motion effect on radial velocity.

**4)** p. 7, 1st paragraph: It should be stated, how the background noise signal is obtained, e.g. via the recorded signal after a
15   sufficiently long laser travel time, or via a separate measurement w/o laser pulse emission. Do the authors see an advantage of their SNR definition over the one from Banakh et al. 2013?

R: The SNR in this study is defined as the ratio of the peak value of FFT spectral signal in each range bin to the Root-Mean-Square (RMS) of background noise signal. Figure 1 shows the array of the spectral $S(l\Delta f; k\Delta R)$, where $l = 0,1,2,3,...,L-1$
20   is the spectral channel number and $L = 100$. In this case the frequency resolution $\Delta f \approx 0.98$ MHz and the corresponding velocity resolution is $\Delta V = 0.76$ ms$^{-1}$. The bandwidth $B_{100} = (L-1)\Delta f = 97.68$ MHz, and the corresponding radial velocity measurement range is $\pm 37.5$ ms$^{-1}$. Figure 1a shows the last 10 range gates raw array of spectral in green line. We estimate the averaged background noise spectrum

$$\overline{S}_N(l\Delta f) = \frac{1}{10} \sum_{k=94}^{103} S(l\Delta f; k\Delta R) \qquad (8)$$

Substracting the background noise spectral $\overline{S}_N(l\Delta f)$ from the raw spectral array $S(l\Delta f; k\Delta R)$, the unnoisy array of spectral
25   $S(l\Delta f; k\Delta R)$ can be obtained and shown in red line in Fig. 1. The peak value index $l_{peak}$ from the $S(l\Delta f; k\Delta R)$ can be firstly obtained and thus the absolute signal power $P_s(k\Delta R)$ at various ranges $k\Delta R$ can be represented as:

$$P_s(k\Delta R) = S(l_{peak}\Delta f; k\Delta R) - \frac{1}{12}\left(\sum_{l_{peak}-20}^{l_{peak}-15} S(l\Delta f; k\Delta R) + \sum_{l_{peak}+15}^{l_{peak}+20} S(l\Delta f; k\Delta R)\right) \tag{9}$$

Replacing integration by summation and taking into account that the zero velocity point in one channel is $l_{zero} = 50$, we estimate the noise power $P_N$ as

$$P_N = \frac{1}{10}\sum_{k=94}^{k=103}\sqrt{\frac{1}{21}\sum_{l=l_{zero}-10}^{l_{zero}+10}\hat{S}_N(l\Delta f; k\Delta R)^2} \tag{10}$$

Finally, we obtain the range profile of the $SNR(k\Delta R)$ using the equation

$$SNR(k\Delta R) = 10\log_{10}\left(\frac{P_s(k\Delta R)}{P_N}\right) \tag{11}$$

[Figure]

Figure 1: The CDL measured array of the FFT spectra (a) the last 10 range gates spectra for background noise spectrum estimation (b) the 1st – 5th range gates (150 m – 270 m, range resolution is 30 m) spectrum.

The SNR from Banakh et al. 2013 is defined as the ratio of the averaged heterodyne signal power $P_s$ to the average detector noise power $P_n$ in a 50-MHz bandwidth. The power $P_s$ and $P_n$ are integrals of the spectral densities $S_s(f)$ and $S_n(f)$, respectively, in frequency f within a band of width $B_{50}$, that is:

$$P_s = \int_{B_{50}} S_s(f)df \tag{5}$$

$$P_n = \int_{B_{50}} S_n(f)df \tag{6}$$

Comparing the definition from Banakh et al. 2013, the SNR in this paper is simpler and also indicates the CDL detection capability, data accuracy and atmospheric tracer particle relative intensity. In this sense, the SNR threshold value in this paper is higher than the one in previous studies (Banakh et al. 2013; Achtert et al 2015) for the same signal power spectrum.

**5)** p. 7, L24: It should be described how the wind fluctuations are determined. Is it the standard deviation of wind measurements of higher temporal resolution (resolution?) during the 10 min.? Why are bars shown only for part of the profile in Fi.g 4 and 5? Is it smaller than a specific value below 1.4 km in Fig. 4? Do the fluctuations represent instrument noise or atmospheric fluctuations? What could be the reason that there are higher fluctuations in the layer of 1.4-1.6 km in Fig. 4?

R: The black line indicates the mean measurement by CDL during the 10-min period, and the red line shows the result which is obtained from simultaneous radiosonde data. The blue bars represent the standard deviation of CDL wind measurement from the 2-min temporal resolution results during the chosen analyzed period, representing the atmospheric fluctuations.

The standard deviation of wind speed and direction below 1.4 km are less than 0.5 m/s and 5°, respectively, showing that the atmospheric condition is relative stable below 1.4 km. While there are higher fluctuations in the height of 1.4 – 1.6 km. The higher SNR in the layer of 1.4 – 1.6 km shown in Fig. 5a implies the existence of cloud or aerosol layer, more active and complex atmospheric movement in this layer may results in higher fluctuations.

**6)** p. 7, L27: Same question related to the method to determine the STD for the angles. Determined from the variability during the 10 min using raw data with of temporal resolution of xx s?

R: I didn't explain it clearly. The related description has added to the manuscript, and it is shown below:

It is noted that the standard deviation of the angles is determined from the variability during the 10 min period using N=1200 raw data with temporal resolution of 0.5 s, which is shown in Fig. 7b.

**7)** p.7, L28: Which SNR threshold was used here?

R: The SNR threshold in this study is 8 dB. The reason why SNR threshold is 8 dB has been analyzed in Sect. 3.3.

**8)** p.8, L26, last sentence: What is a "multipath effect"? This should be clarified. Also the difference in radiosonde and lidar location should be stated quantitatively. What is the difference in mean wind speed and direction between radiosonde and lidar above 1 km? Can a lidar instrumental effect excluded to explain the difference? I am not convinced that it is only colocation.

R: The difference in mean wind speed and direction between radiosonde and CDL above 1 km is about 3.4 $ms^{-1}$ and $15.2°$, respectively, showing significant discrepancy. On the one hand, the random error of the corrected CDL estimation of the wind due to the low SNR shown in Fig. 6a contributes to this discrepancy. On the other hand, according to the recorded information, the mean heading angle and cruising speed of the ship is $75.86°$ and 4.84 $ms^{-1}$, respectively, and the mean wind speed and direction above 1 km is $255°$ and 18.4 $ms^{-1}$, respectively. Since the drift of radiosonde is affected by atmospheric wind, and turbulence perturbation and the CDL detection volume is changing during cruising observation, the result discrepancy between radiosonde and CDL caused by different observation location, also called the multipath effect, is larger with increasing height.

**9)** p.8 and Fig. 6: I would propose to plot the radiosonde on the x-axis and the lidar on the y-axis and also perform the linear least square fit with these coordinates. I consider the radiosonde as more accurate and the usual linear LSF procedures assume that the x-parameter is without errors (minimization of vertical differences). I also consider the criteria of excluding data with 1*SD as too strict. Only gross outliers – deviating from a Gaussian distribution – could be excluded. This would typically result in a criteria of >3*SD.

It needs also to be stated, how many data-pairs were excluded from the statistical comparison in order to judge the numbers of gross outliers. Also the SD typically refers to the SD of the difference (lidar-radiosonde). I am wondering how the SD of the lidar data ydata was obtained here. It is clear that the statistical parameters for bias, SD, R, and RMSE need to be calculated without a rigorous excluding of the data (with 1*SD). This point needs to be revisited and clarified.

R: I agree with your suggestion, the radiosonde on the x-axis and the lidar on the y-axis has been used in the manuscript. Figure 1 shows the distribution of difference (lidar-radiosonde) and fitted Gaussian distribution. The total number of wind speed and direction dataset is 1062 and 951, respectively. The 1*SD, 2*SD, 3*SD are plotted in red, black and blue dotted-line, respectively. The SD is the standard deviation of the difference of (lidar-radiosonde). It can be seen from figure 1 that the criteria of excluding data with 2*SD is more reasonable for gross outliers. Figure 2-4 shows the comparison of lidar and radiosonde using different criteria. The excluded data-pair using the different criteria are listed in Table 1 below:

Table 1. Excluded data-pair and corresponding % using different criteria

|       | Excluded wind speed data-pair (%) | Excluded wind direction data-pair (%) |
|-------|-----------------------------------|---------------------------------------|
| 3*SD  | 14 (1.3%)                         | 12 (1.3%)                             |
| 2*SD  | 62 (6%)                           | 56 (5.9%)                             |
| 1*SD  | 252 (21%)                         | 225 (24%)                             |

The statistical parameters for bias, SD, R, and RMSE after data quality control with different criteria are shown in figure 2-figure 4.

[Figure]

Figure 1: Distribution of difference (lidar-radiosonde) (a) wind speed (m/s) (b) wind direction (°)

[Figure]

Figure 2: Comparison of (a) wind speed and (b) wind direction between CDL and radiosonde data using 3*SD threshold

[Figure]

Figure 3: Comparison of (a) wind speed and (b) wind direction between CDL and radiosonde data using 2*SD threshold

[Figure]

Figure 4: Comparison of (a) wind speed and (b) wind direction between CDL and radiosonde data using 1*SD threshold.

**10)** p.9, ch. 3.2 and Fig. 7: The dots for MABL height are shown for the first 1/3 of Fig. 7 in a region of SNR around 10, where no obvious gradients can be seen, whereas for the second 2/3 it is more in the region between 10 dB (light blue) and 0 dB (dark blue). Please check and comment. Is there a reference about the ABL height determination using the first negative gradient?

R: We didn't explain it clearly. The MABL height has been retrieved and compared using different instruments such as the CDL, radiosonde, and CL31 ceilometer during this campaign (Wang et al., 2016). Many papers have discussed the use of backscatter signal of Lidar for boundary layer height estimation, assuming that the boundary layer has higher aerosol concentrations than the free troposphere above. In this paper, the SNR, representing the relative aerosol backscatter profiles, were used and two common methods includes thresholding SNR to determine MABL height (Melfi et al. 1985) and finding the height of the first strong negative gradient (White et al. 1999; Hennemuth and Lammert 2005 ) in SNR. Figure 1a shows the Time-Height-Intensity of SNR and the retrieved MABL height marked with black and red solid circles. The radiosonde data during 17:34 LST 14 May 2014 and corresponding MABL height using the gradient of potential temperature and relative humidity are also shown in Fig. 2. It can be seen that diurnal variation of MABL height is less obvious within 1.0 km - 1.5 km, consistent with the mixing layer height retrieved from the radiosonde potential temperature and relative humidity profile.

The related references have been added to the manuscript:

1. Hennemuth, B., and Lammert, A.: Determination of the atmospheric boundary layer height from radiosonde and lidar backscatter, Boundary-Layer Meteorol., 120(1), 181-200, 2006.

2. Menut, L., Flamant, C., Pelon, J., and Flamant, P. H.: Urban boundary-layer height determination from lidar measurements over the Paris area, Appl. Opt., 38(6), 945-954, 1999.

3. Wang, D., Song, X., Feng, C., Wang, X., and Wu, S.: Coherent Doppler Lidar Observations of Marine Atmospheric Boundary Layer Height in the Bohai and Yellow Sea, Acta Opt. Sin., 35(A01), 1-7, 2015.

4. White, A. B., Senff, C. J., and Banta, R. M.: A comparison of mixing depths observed by ground-based wind profilers and an airborne lidar, J. Atmos. Oceanic. Technol., 16(5), 584-590, 1999.

[Figure]

Figure 1: Example measurement from 07:33 to 15:29 LST 14 May 2014: (a) Time-Height-Intensity of SNR and retrieved MABL height using SNR threshold and gradient method (black and red solid circles, respectively). (b) Time-Height-Intensity of vertical velocity after attitude correction.

[Figure]

Figure 2: Radiosonde profiles of (a) potential temperature (K) (b) the gradient of potential temperature (c) relative humidity (%) and (d) the gradient of relative humidity at 12:00, LST 14 May 2014. The horizontal red and green lines in (b) and (c) stand for MABL height retrieved from potential temperature and relative humidity, respectively.

**11)** p.11, L5: I consider an error of only 0.1_ for the ship heading as very small. Is this justified by the hard-target measurements?

R: The error of 0.1 for ship heading comes from the accuracy of Global Navigation Satellite System.

**12)** p.11, L8: What quantity is derived in eq. (14) in comparison to eq (13); Both are called "bias" LOS_N but eq. (14) with a "N'". Text should clearly state, the difference. What eq. (13 or 14) is then used in the estimates for the bias (eq. 17 and 18)?

R: The $\vec{V}_{LOS}$ is the LOS velocity in Earth coordination system with elevation $\theta_g$. The $\vec{V}'_{LOS}$ is the LOS velocity in Earth coordination system with elevation $\theta_0 = 60°$ in this study. The relationship between $\vec{V}_{LOS}$ and $\vec{V}'_{LOS}$ is $\vec{V}'_{LOS} = \vec{V}_{LOS} \cos\theta_0 / \cos\theta_g$. The bias of $\vec{V}_{LOS}$ is derived using eq (13) and the bias of $\vec{V}'_{LOS}$ is affected by $\vec{V}_{LOS}$ and $\theta_g$ according to the error propagation theory, as shown in eq. (14). The estimation for horizontal wind bias shown in eq. 17 and 18 are related to $bias_{b1}$ and $bias_{b2}$. According to $b_1 = (\vec{V}'_{LOS\_N} - \vec{V}'_{LOS\_S})/\cos\theta_0$ and $b_2 = (\vec{V}'_{LOS\_E} - \vec{V}'_{LOS\_W})/\cos\theta_0$, the bias of $\vec{V}'_{LOS}$ will be used.

**13)** p.11, eq. 15/16: These eq. could be moved to ch. 2 after eq (10), because it deals with u, and v retrieval and not with error estimates as in ch. 3.3.

R: Thanks for your suggestion, the eq 15-16 has moved to ch.2 after eq.10.

**14)** p.11, L13: Here it is stated, that the lidar pointing angles are very small (and assumed to be perfect), but on p.12, L2 it is stated the errors are dominated by ship velocity and lidar pointing errors. This is in contradiction.

R: I did not explain it clearly. Because of the requirement for small bias in the radial velocity measurements, the error in the laser beam direction must be very small and one can assume perfect knowledge of the coefficient $a_i$.

The dominant source of bias of the horizontal velocity estimates come from the biases of the radial velocity estimates, which are determined by the error in the ship velocity $\vec{V}_{ship\_horizontal}$, $\vec{V}_{ship\_vertical}$ and heading angle $\psi$ and lidar pointing knowledge errors $\Delta\varphi$ and $\Delta\theta$ (see Eq. 13).

**15)** p.12, L6ff: Here the method of obtaining the random error is described ("In this case, a"). But no resulting spectrum is shown in Fig. 9. This needs to be added or reformulated.

R: Various methods of estimating the magnitude of the random error of Doppler Lidar velocity measurements have been introduced (Frehlich 2001). The measurement of error from velocity spectrum were used in this paper. A 50 % window overlap factor, a Hamming window is used in order to reduce the leakage in the spectra. A zero-padding of the missing values were applied to each window for each spectrum calculation to improve the frequency resolution. The constant high-frequency region of velocity spectrum higher than 0.2 Hz, shown in figure below, represents uncorrelated random error contribution, which is departing from the Kolmogorov's -5/3 law. The random error of vertical wind velocity is estimated as the standard deviation of the measured signal after high-pass filter.

[Figure]

Figure 1: Power spectral density P(f) without and with Hamming window for the CDL measured vertical speed between 15:52 and 16:02 LST on 09 May and for an altitude of 1495 m (blue and black solid line, respectively). The expected spectral behaviour according to the Kolmogorov's−5/3 law (pink solid line), the noise frequency threshold (red dotted line) and the derived noise floor for the CDL ( green dotted line) are shown.

**16)** P12: L12: Are you sure that it is an elevated aerosol layer and not a cloud, which provides the high SNR around 1.5 km?

R: Thanks for your suggestion. Actually, we cannot judge whether it is an elevated aerosol layer or cloud only using the SNR intensity signal. We searched the recorded information from Vaisala CL31 ceilometer software screenshot, as shown below:

[Figure]

Figure 1: CL31 ceilometer software real-time results on 14 May 2014.

In this figure, the candidate boundary layer height and cloud base height can be marked with black squares and white squares, respectively. The high SNR around 1.5 km during 07:33 – 08:40 LST is an aerosol layer, not cloud layer. The description has been corrected in the manuscript.

**17)** p.12, L23: speckle-induced phase noise is not discussed in Achtert et al. 2015. Another reference needs to be provided
R: The related references have been added to the manuscript.

1. Frehlich, R.: Effects of wind turbulence on coherent Doppler lidar performance, J. Atmos. Oceanic. Technol., 14(1), 54-75, 1997.
2. Frehlich, R.: Estimation of velocity error for Doppler lidar measurements, J. Atmos. Oceanic. Technol., 18(10), 1628-1639, 2001.

**18)** p.13 Summary: The limitations of the approach in comparison to existing systems need to be mentioned in the summary. Also I am missing an outlook about future algorithm or hardware improvements or future deployment during ship cruises.

R: The limitation of the approach has been discussed in General and Major Comments Question 1.

The outlook has been described in General and Major Comments Question 7.

**19)** p.13, L14: The number for the bias and the STD from the statistical comparison of all radiosondes should be stated here.

R: The total number of wind speed and direction dataset for comparison is 1062 and 951, respectively.

**20)** Ref. Liu et al. 2010: More details should be provided for this reference, which is not really accessible, or the reference should be removed or replaced. Also Achtert et al. (2015) provide these transformations.

R: In Liu et. al 2010 paper, a mobile Doppler lidar had been developed for 3D wind measurements by Ocean University of China. In order to further improve the mobility of the mobile Doppler lidar for lidar calibration and validation, both GPS and inertial navigation system were integrated on the vehicle for performing measurements during movement. The modifications of the system and the results of the moving measurements were presented. This work simplifies the construction of the mobile Doppler system and makes the lidar more flexible for ground-based wind measurements and validation with the ADM-Aeolus spaceborne Doppler lidar.

**21)** Fig. 1: An additional Figure should be shown of the ship to illustrate the location of the CDL on the ship and possible disturbances of the flow.

R: Revised

**22)** Fig. 1: The location of the INS on the CDL should be indicated in the Figure.

R: Revised

**23)** Fig. 2: The symbols used for the angles pitch, roll, yaw should be placed also in the Figures.

R: Revised

**24)** Fig. 7: the legend within Fig. 7b is too small

R: Revised

%%==============================================================================%%

Editorial: A large number of editorial comments were directly added to the PDF-Version of the manuscript. In addition the term "et al" needs to be replaced by "et al.". The manuscript needs thorough proof-reading after revision.

**1)** P2 L27: "High Resolution Doppler Lidar (HSRL)" needs to be corrected also at other places

R: Revised

**2)** P4 L25: Check style file or other papers, if "," or ";" is needed to separate coordinates.

R: Revised, $X_g, Y_g, Z_g$

**3)** P5 L3: This is not clear: why is the azimuth angle changing, when looking downward.

R: From the top view, the $\varphi_s$ increases in a clockwise direction during 4-DBS mode operation.

**4)** P5 L4: This must be the Xs-Zs plane.

R: we have checked the definition, it is the Xs-Ys plane.

**5)** P5 L8: Here the indices "g" are missing.

R: the "g" has added to the manuscript

**6)** P6 L10: "homogenous" flow instead "cellular"

R: revised

**7)** P6 L17: N, S, E and W, not uppercase style.

R: revised

**8)** P8 L15: sentence not completed: "What's more, the fluctuation in wind speed and direction above 1 km is more severe."

R: What's more, the fluctuation in wind speed and direction above 1 km is more severe than the results below 1 km.

**9)** P9 L3: "the coefficient of determination of 0.96", I assume this value is R^2. I consider it sufficient to provide R.

R: Yes, the coefficient of determination of 0.96 is R^2, the description is delated in the manuscript.

**10)** P20 Fig 7. This text is too small; also the quantities should be plotted with differetn y-scales to see more details.

R: revised, please see figure 11 in the revised vision.

**11)** P23 Table 1 could be replaced by power consumption.

R: revised

**12)** P24 Table 2 are these numbers accuracy and precision?

R: Yes, revised.

**13)** P24 Table 3: I would consider only 2 significant digits for normalized RMSE and direction values as sufficient, e.g. 4.6 (instead 4.55) or 4.3 (instead 4.27)

R: revised.

**14)** In addition the term "et al" needs to be replaced by "et al.". The manuscript needs thorough proof-reading after revision.

R: revised.

**3. Reviewer 3**

**Summary:**

This manuscript describes a study that is relevant to Atmospheric Measurement Techniques. The authors describe procedures and measurement performance of a coherent Doppler wind lidar (CDWL) from ship. The manuscript proposes an algorithm to compensate for error of wind measurement due to the motion of the ship and provides contributions for lidar communities using shipborne applications. The manuscript has some issues that need clarification. There are numerous specific comments. A major revision of manuscript is needed before it can be accepted for publication.

**General comments:**

**1)** Although details of a CDWL WindPrintS4000 are not described in the previous papers, in my opinion, details of the CDWL are not described well. The authors use an AOM for the heterodyne detection and a FPGA for FFT analysis. But there is no information about the sampling frequency and points used for FFT. The information is related to range-resolution for the time-domain, frequency-resolution for the frequency-domain and observable wind speed range. In the manuscript, bandwidth of 50MHz is used for data processing. Do you use a FPGA operating at a sampling frequency of 100MHz? Is it correct? In the previous paper, a AOM of 80MHz is used for the heterodyne detection. Therefore, frequency range of 60-100 MHz at center of 80MHz is detection range to determine LOS wind speed. 20MHz corresponds to be LOS wind speed of 15.5 m/s. ±50 m/s is "speed measurement range" shown in the previous paper. It is not consistent each other. It is puzzled to me. I might be missing something...and if so, please describe technical details and other aspects of the CDWL for better understanding the manuscript.

R: The SNR in this study is defined as the ratio of the peak value of FFT spectral signal in each range bin to the Root-Mean-Square (RMS) of background noise signal. Figure 1 shows the array of the spectral $S(l\Delta f; k\Delta R)$, where $l = 0, 1, 2, 3, ..., L-1$ is the spectral channel number and $L = 100$. In this case the frequency resolution $\Delta f \approx 0.98$ MHz and the corresponding velocity resolution is $\Delta V = 0.76$ ms$^{-1}$. The bandwidth $B_{100} = (L-1)\Delta f = 97.68$ MHz, and the corresponding radial velocity measurement range is $\pm 37.5$ ms$^{-1}$. The elevation angle for 4-DBS is normally set as $60°$, so the detectable maximum horizontal wind speed is $\pm 106$ ms$^{-1}$.

**2)** Definition of SNR is given in the manuscript. SNR=0dB means the signal power equals to noise power (NEP). Is it correct? Do you mean that the minus values are bad estimates? Minus values of SNR are shown in Figures 7 and 9. Why? Please describe details and derivation of SNR and search procedure by adding explanation sentences and figure.

R: The SNR in this study is defined as the ratio of the peak value of FFT spectral signal in each range bin to the Root-Mean-Square (RMS) of background noise signal. Figure 1 shows the array of the spectral $S(l\Delta f; k\Delta R)$, where $l = 0,1,2,3,...,L-1$ is the spectral channel number and $L = 100$. In this case the frequency resolution $\Delta f \approx 0.98$ MHz and the corresponding velocity resolution is $\Delta V = 0.76$ ms$^{-1}$. The bandwidth $B_{100} = (L-1)\Delta f = 97.68$ MHz, and the corresponding radial velocity measurement range is $\pm 37.5$ ms$^{-1}$. Figure 1a shows the last 10 range gates raw array of spectral in green line. We estimate the averaged background noise spectrum

$$\overline{S}_N(l\Delta f) = \frac{1}{10}\sum_{k=94}^{103} S(l\Delta f; k\Delta R) \tag{8}$$

Subtracting the background noise spectral $\overline{S}_N(l\Delta f)$ from the raw spectral array $S(l\Delta f; k\Delta R)$, the unnoisy array of spectral $S(l\Delta f; k\Delta R)$ can be obtained and shown in red line in Fig. 1. The peak value index $l_{peak}$ from the $S(l\Delta f; k\Delta R)$ can be firstly obtained and thus the absolute signal power $P_s(k\Delta R)$ at various ranges $k\Delta R$ can be represented as:

$$P_s(k\Delta R) = S(l_{peak}\Delta f; k\Delta R) - \frac{1}{12}\left(\sum_{l_{peak}-20}^{l_{peak}-15} S(l\Delta f; k\Delta R) + \sum_{l_{peak}+15}^{l_{peak}+20} S(l\Delta f; k\Delta R)\right) \tag{9}$$

Replacing integration by summation and taking into account that the zero velocity point in one channel is $l_{zero} = 50$, we estimate the noise power $P_N$ as

$$P_N = \frac{1}{10}\sum_{k=94}^{k=103}\sqrt{\frac{1}{21}\sum_{l=l_{zero}-10}^{l_{zero}+10} \hat{S}_N(l\Delta f; k\Delta R)^2} \tag{10}$$

Finally, we obtain the range profile of the $SNR(k\Delta R)$ using the equation

$$SNR(k\Delta R) = 10\log_{10}\left(\frac{P_s(k\Delta R)}{P_N}\right) \tag{11}$$

[Figure]

Figure 1: The CDL measured array of the FFT spectra (a) the last 10 range gates spectra for background noise spectrum estimation (b) the 1st – 5th range gates (150 m – 270 m, range resolution is 30 m) spectrum.

5  The SNR from Banakh et al. 2013 is defined as the ratio of the averaged heterodyne signal power $P_s$ to the average detector noise power $P_n$ in a 50-MHz bandwidth. The power $P_s$ and $P_n$ are integrals of the spectral densities $S_s(f)$ and $S_n(f)$, respectively, in frequency f within a band of width $B_{50}$, that is:

$$P_s = \int_{B_{50}} S_s(f) df \qquad (5)$$

$$P_n = \int_{B_{50}} S_n(f) df \qquad (6)$$

Comparing the definition from Banakh et al. 2013, the SNR in this paper is simpler and also indicates the CDL detection
10  capability, data accuracy and atmospheric tracer particle relative intensity. In this sense, the SNR threshold value in this paper is higher than the one in previous studies (Banakh et al. 2013; Achtert et al 2015) for the same

**3)** It is also necessary to describe here the statistical process of these measures. How did you calculate LOS wind speed error and bias? How many radiosondes did you launch? What is the vertical resolution of the radiosonde? How do you interpolate
15  to the CDWL data to for compare with the radiosonde? Ex. vertical resolution of the CDWL is 60m, while the vertical resolution of the radiosonde is 30m. The tow data points measured by the radiosonde are used for comparison with the CDWL. Or, one data averaged using two data points are used for that. Are data in the altitude range between 150m and 4000m used for the statistical comparison show in the Figure 6. A question comes for the difference between number N of 990 shown in Figure 6 and number points (wind speed: 84, 106...65; wind direction: 89,93...88) shown in Table 4. Why are the numbers
20  used for the wind speeds and wind directions are different? Please add explanation related to spatial and temporal difference between the DBS and the radiosonde measurements.

R: The bias of the LOS wind is calculated using Eq. 12-15 in the revised manuscript, and it is based on the error propagation theory. The error of LOS wind at specific azimuth and elevation angle is difficult for shipborne measurement since the ship

motion results in measurement from various directions. It is different from the error analysis of vertical velocity. The corrected vertical velocity is the velocity from zenith stare direction. Thus the error of vertical velocity can be obtained from the time series of corrected vertical velocity using frequency spectrum analysis.

5 In order to assess the accuracy of the shipborne lidar wind measurement, a comparison of the lidar measurement and 11-radiosonde dataset during the experiment has been made. It is noted that the range resolution of lidar in this study is 30 m, and the corresponding vertical resolution with elevation angle of $60°$ is about 26 m. The vertical resolution of radiosonde is 10 m. During the comparison, the wind profile of radiosonde should be interpolated to the common height grid with finer resolution of 2 m firstly, and then the data point closest to height point of lidar will be chosen for comparison. The measurement range

10 in this study is between 150 m and 3240 m (corresponding to the 104th range bin), thus the altitude range between 130 m and 2806 m are used for the statistical comparison shown in the Fig. 8.

The dataset for comparison in Fig.8 in the revised manuscript excludes the data where $|ydata - xdata| > 2*SD$. The $ydata$ and $xdata$ is the Lidar and corresponding radiosonde data, respectively, and SD represents the standard deviation of the

15 difference of $ydata - xdata$. According to the distribution of difference of $ydata - xdata$ and fitted Gaussian distribution shown below, the criteria of excluding data with $2*SD$ is reasonable for gross outliers. The excluded data-pair number and proportion is 62, 6% for wind speed and 56, 5.9% for wind direction, respectively. The number used for the wind speeds and wind directions are different since the $|ydata - xdata| > 2*SD$ in wind speed and direction comparison is different from each other.

[Figure]

Figure 1: Distribution of difference (lidar-radiosonde) (a) wind speed (m/s) (b) wind direction (°)

**4)** How did you determine misalignment and angle between the ship and laser beam axes? Although explanation sentences using a hard target are descried in the manuscript, technical details and other aspects of this are not described well. More details should be provided.

R: In our system, the inertial navigation system is rigidly mounted on the base of the scanner, instead of the deck of the ship, to keep constant relative angles with reference to the transmitting laser beam. It records the Lidar motion angles including pitch, roll, laser beam azimuth and elevation, thus the recorded attitude information is the exact Lidar itself feature in Lidar coordinate system. After installation, a hard target calibration is firstly performed to determine the misalignment between the ship and laser beam axes. Specifically, the buildings near the wharf where there is no occlusion issue between the CDL and the candidate buildings can be chosen as the hard target. As shown in Fig.1, when the laser beam direction points to the hard target, the azimuth angle $\varphi_{lidar}$ in Lidar coordinate system is recorded, meanwhile the azimuth angle $\varphi_g$ in Earth Coordinate System can be obtained using the Google Earth software if the exact longitude and latitude of hard target is determined. According to the ship heading angle $\psi$, we can get the azimuth angle $\varphi_s = \varphi_g - \psi$ between ship heading and the hard target in Ship Coordinate System. So far, the misalignment angle between the ship and laser beam axes $\Delta\varphi = \varphi_s - \varphi_{Lidar}$ can be corrected using the geometrical relationship between these three angles. And then the standard ship attitude definition can be determined based on the relationship between Lidar and ship coordinate system, which will be used in the following ship motion correction process. It can be seen that there exists no laser direction error determined by misalignment between the ship and laser beam axes since the Lidar is considered to be relative static during field experiment.

[Figure]

Figure 1. The overhead view of Lidar, ship and Earth coordinate system and corresponding hard target calibration.

**Specific Comments:**

**1)** P. 2 line 22, "wind speed". Do you mean "wind vector"?

R: Yes, revised

**2)** P. 3 line 4, "the CDL". a CDL

R: revised

**3)** P. 3 line 4, "vertical wind". horizontal wind

R: revised

**4)** P. 3 line 11, "In order to ...the Yellow sea". What is main purpose of the experimental investigation? Scientific or Engineering?

R: The main objective and further plan is described in the revised version Introduction and Summary, see below:

Introduction part:

As one of the main objectives, the CDL was deployed on the ship in this campaign to demonstrate the feasibility of the algorithm-based attitude correction method. The obtained accurate three-dimensional wind information can provide significant preparation for further studies on characteristics of dynamics and thermodynamics in the MABL and turbulence flux exchange over sea surface. In addition to CDL, as another important part of this campaign, a High Spectral Resolution Lidar (HSRL) and a Vaisala CL31 ceilometer were also deployed on the ship platform in order to detect MABL height spatial-temporal evolution and to retrieve the aerosol and cloud optical characteristics such as extinction coefficient and backscatter ratio and so forth. It will help us to understand the complex behaviour of MABL and the aerosol cloud forcing characteristics over sea region and the impact on climate change. This paper focuses on CDL performance and gives a thorough analysis of the attitude correction for lidar velocity measurement.

Summary part:

Overall, combining a CDL with attitude correction system and accurate motion correction process as presented here forms a reliable and autonomous set-up that could be placed on mobile platform to provide more detailed, higher spatial and temporal resolution view of three-dimensional wind field information. It will be further validated and improved under different sea conditions using CFD model simulation in further field campaign. More specific studies are being carried out or prepared, including atmospheric turbulence characteristics statistics and multi-scale wind field observation in MABL, wind turbine wake and atmospheric turbulence interaction over offshore wind power field (Wu et al., 2016; Zhai et al., 2017), mass transport and

flux analysis in MABL with combination of CDL and Multi-wavelength Polarization Raman Lidar (Wu et al., 2016), and the forthcoming ADM-Aeolus wind data validation over China Sea in 2018.

**5)** P.3 line 30, "200ns". Is the number for 150μJ at 10 KHz?

5  R: The pulse width produced by the modulation is adjustable from 100 ns to 400 ns. In this study, the pulse width of 200 ns, corresponding to the range resolution of 30 m, is used. The pulsed energy is approximately 150 μJ and the pulse repetition frequency is 10 kHz.

**6)** P.4 line 2, "a proper". Do you mean "high"?

10  R: Yes, revised

**7)** P.4 line 9, "Yellow Sea". "the Yellow Sea". There are the same expressions in the manuscript. Please check in the manuscript
R: Revised

15  **8)** P.4 line 13-15, "The inertial navigation system is ...laser beam". How did you confirm to keep the constant relative angle?
R: I didn't explain it clearly.
"The inertial navigation system is rigidly mounted on the base of the scanner, instead of the deck of the ship, to keep constant relative angles with reference to the lidar coordinate system.

20  **9)** P.4 line 16-17, "Hard target calibration". How did you conduct out the hard target calibration? Please describe details about it and statistical results (bias and error)
R: The specific description can be seen in General comments question 3. Generally, the buildings near the wharf where there is no occlusion issue between the CDL and the candidate buildings can be chosen as the hard target. As shown in Fig.1, when the laser beam direction points to the hard target, the azimuth angle $\varphi_{lidar}$ in Lidar coordinate system is recorded, meanwhile

25  the azimuth angle $\varphi_g$ in Earth Coordinate System can be obtained using the Google Earth software if the exact longitude and latitude of hard target is determined. According to the ship heading angle $\psi$, we can get the azimuth angle $\varphi_s = \varphi_g - \psi$ between ship heading and the hard target in Ship Coordinate System. So far, the misalignment angle between the ship and laser beam axes $\Delta\varphi = \varphi_s - \varphi_{Lidar}$ can be corrected using the geometrical relationship between these three angles. And then the standard ship attitude definition can be determined based on the relationship between Lidar and ship coordinate system, which

30  will be used in the following ship motion correction process. It can be seen that there exists no laser direction error determined by misalignment between the ship and laser beam axes since the Lidar is considered to be relative static during field experiment.

[Figure]

Figure 1. The overhead view of Lidar, ship and Earth coordinate system and corresponding hard target calibration.

**10)** P.4 line 19-20, "laser direction". Do you mean "laser beam direction"? What do you mean "between the ship and..."? Please explain it and show the axes in Figure 2.

R: Yes, the "laser direction" means "laser beam direction". The ship and Lidar coordinate system are shown in fig.2 in revised version with red and green arrowed lines, respectively. "between the ship and …" mean the the laser direction error determined by misalignment between the ship and Lidar is negligible since the Lidar is considered to be relative static compared with ship during the field experiment.

**11)** P.5 line 3, "the transmitting laser path". Do you mean "laser beam direction"?

R: Yes, revised

**12)** P.5 Eq. 3, (x, y, z) ->(xg, yg, zg)

R: Revised, see Appendix A

**13)** P.5 line 9, "Where" -> "where"

R: Revised

**14)** P.5 line 10, "Eq. (2)-(3)" -> "Eqs. (2) and (3)"

R: Revised

**15)** P.5 Eq. 4, ( y/x) ->(yg / xg)

R: Revised

**16)** P.5 Eq. 5, z -> zg

R: Revised

**17)** P.7 line 3-4, please describe the definition of SNR using a figure. Does the definition of SNR have minus value?

R: The specific description of SNR can be seen in General comment question 1.

**18)** P.7 line 16-17, please give observation time to get each LOS wind speed profile.

R: Revised, the temporal resolution of radial velocity is 0.5 s.

**19)** P.7 line 24, How do you determine the wind measurement fluctuation?

R: In this study, the horizontal wind profile with 2-min temporal resolution will be retrieved for vertical velocity correction. Basically, the LOS velocities from $N$, $S$, $E$, and $W$ direction after SNR quality control during the chosen 2-min interval are collected firstly. Then the procedure of filtration of reliable estimates of each radial velocity based on SNR threshold is used to obtain "good" speed estimates. The selected radial velocities and corresponding ship condition information in each radial direction are averaged and the averaged ship condition will be used for the removal of platform velocity effect. Finally, the horizontal with 2-min temporal resolution can be retrieved using modified 4-DBS mode. The vertical wind measurement has a temporal resolution of 0.5 s, the horizontal wind whose retrieved time is closest to vertical wind measured time will be used for vertical velocity correction.

The blue bars shown in Fig. 5 and 6 represent the standard deviation of CDL wind measurement from the 2-min temporal resolution results during the chosen analyzed period, which can effectively represent the atmospheric fluctuations.

**20)** P.7 line 28, "SNR threshold". Please add the threshold value.

R: Revised. Data quality control based on SNR threshold is used to remove the spikes higher than 2.4 km . The SNR threshold in this study is 8 dB and the reason has been analyzed in Sect. 3.3.

**21)** P.8 line 26, "measurement". "measurements"

R: Revised.

**22)** P.8 line 27, "multipath". I do not understand it. Please add explanations.

R: Since the drift of radiosonde is affected by atmospheric wind and turbulence perturbation, and the CDL detection volume is changing during cruising observation, the result discrepancy between radiosonde and CDL caused by different observation location, also called the multipath effect, is larger with increasing height.

**23)** P.11 line 1, "shipborne-based". "ship-based" or "shipborne".

R: Revised.

**24)** P.11 line 9-10, "assuming that the wind field has a constant horizontal and vertical velocity". Is the assumption always reasonable? What is the spatial and temporal scale of wind field when the assumption is reasonable.

R: Whether the assumption is reasonable or not depends on particular investigated process, land surface condition and atmospheric stratification stability. Specifically, under the assumption of homogeneous flow with little turbulence which would lead to a smooth sinusoidal behavior in the VAD scan, it can be expected that 4-DBS mode should be sufficient, along with one measurement in the vertical. It is faster and simpler both in the hardware and in the data evaluation algorithm, but lacks the goodness-of-fit information as a measure for the reliability of the results (Weitkamp 2005). This shortcoming is partially compensated by information about the temporal behavior of the data. Therefore, the parameters such as maximum range, range resolution, temporal resolution (or scan rate) need to be set carefully based on a given purpose (Weitkamp, 2005).

In this study, both the ship-induced shift and radial velocity have the same temporal resolution of 0.5 s, and the pulse repetition rate is 10 kHz, which is useful for the detection of small-scale turbulence. The temporal resolution of horizontal wind profile is 2-min, and it is reasonable for the knowledge of background wind field. The temporal resolution of vertical velocity is 0.5 s, which is necessary for atmospheric turbulence characteristics statistics. If the mean wind speed is 5 m/s, the corresponding smallest spatial and temporal scale of wind field for the homogeneous isotropic assumption is 600 m and 2 min, respectively.

**25)** P.11 line 13, "the lidar pointing angle". Do you mean the laser beam direction?

R: Yes, revised.

**26)** P.12 line 6-8, "In this case...2016).". Why do you have to use the Hamming window and zero-padding? Please clarify it. What is the difference between with and without Hamming window and zero-padding?

R: A 50 % window overlap factor, a Hamming window is used in order to reduce leakage in the spectra. A zero-padding of the missing values were applied to each window for each spectrum calculation to improve the frequency resolution. Figure 1 shows the results without and with Hamming window and zero-padding. It can be seen that the P(f) with Hamming window is more smoother and easier to find the constant frequency region for random error estimation.

[Figure]

Figure 1: Power spectral density P(f) without and with Hamming window for the CDL measured vertical speed between 15:52 and 16:02 LST on 09 May and for an altitude of 1495 m (blue and black solid line, respectively). The expected spectral behaviour according to the Kolmogorov's−5/3 law (pink solid line), the noise frequency threshold (red dotted line) and the derived noise floor for the CDL ( green dotted line) are shown.

**27)** P.12 line 16. "Eq.". "Eqs."

R: Revised.

**28)** P.12 line 21-23. "8dB." Why "8dB"? How do you determine the number? Please add explanation sentences. SNR looks the same value at 1 km as at 1.5 km. But different random errors at the altitudes are shown in the Figure 9. Why?

R: It can be seen that in the high SNR region above 8 $dB$, a constant random error range between 0.03 and 0.15 $ms^{-1}$ is found because of the effect of the speckle-induced phase noise (Frehlich, 1997; Frehlich, 2001). At reduced values of the SNR, the errors increase as a result of increasing signal noise, rising to approximately 4 $ms^{-1}$ at an $SNR = 0\,dB$. It is confirmed that the choice of a conservative SNR threshold of 8 $dB$ is robust for data quality control process.

The SNR at 1 km and 1.5 km are 7.3 dB and 7.2 dB, respectively. The Power spectrum density from 1 km and 1.5 km can be seen in figure 1, and figure 2 is the corresponding noise time domain signal. It can be seen that although the SNR has the same level, the standard deviation of velocity can be different because of specific signal.

[Figure]

Figure 1: the power spectrum density at (left) h=1.002 km (right) h=1.495km

[Figure]

Figure 2: Time series of noise time domain signal at (left) h=1.002 km and (right) h=1.495km

**29)** P.12 line 23-25. "At reduced values...0 dB." Please plot results until SNR be 0dB in the Figures 9(a)-9(c). It is important for readers to identify the measurement performance of your CDWL. SNR=0 means the signal power equals to noise power (NEP), which is undetectable a true signal. Random errors would be large. "4 m/s" seems to be small.

R: The figure 13 shown in revised version shows the SNR profile from 0.15 km to 3.105 km. It is noted that the CDL blind
10  area is less than 0.15 km and the maximum detection range is 3.105 km. The minimum of SNR in Figure 1a is 1.08 dB at height of 3.105 km. The SNR=0 means that the peak value of spectrum equals to the mean RMS of background noise spectrum, see Eq.4, thus not representing the signal power equals to noise power. What's more, we didn't explain it clearly, figure 1b shows the random error of vertical velocity where 4 m/s is large for the order of vertical velocity.

[Figure]

Figure 1: The averaged profile of (a) SNR (b) Random error of vertical velocity (c) bias of horizontal wind north-south component (u), east-west component (v) and horizontal wind velocity (h) measured by CDL from 15:52 to 16:02 LST on 09 May, 2014.

**30)** P.14 line 4. "Shipborne wind observation". This manuscript describes an algorithm to compensate for error of wind measurement due to the motion of the ship, not observation. Please modify explanation sentences to insist on the main purpose of the manuscript.

R: Revised. "Shipborne wind observation by a CDL during the 2014 Yellow Sea campaign are carried out to study the structure of the MABL. An algorithm to compensate for error of wind measurement due to the motion of the ship is presented in this paper."

**31)** P.14 line 12. "The correlation...respectively,". Please add details such as date, time, altitude range, and so on.

R: Revised. "In order to assess the accuracy of the shipborne lidar wind measurement, a comparison of the lidar measurement and 11-radiosonde dataset from 09 May 2014 to 19 May 2014 has been made. The total number of wind speed and direction dataset for comparison is 1062 and 951, respectively."

**32)** P.14 line 19-21. "random error...radiosonde data,". Please add explanation sentence about date, time and altitude.

R: Revised. "A case study during 15:52 to 16:02 LST on 09 May 2014 is presented. The height range is from 0.15 km to 3.105 km where the blind area of CDL is less than 0.15 km and the maximum detectable range is 3.105 km. It is found that the random error of vertical velocity is between 0.03 $ms^{-1}$ and 1.2 $ms^{-1}$ and is mainly determined by the SNR, while the bias was less than 0.02 $ms^{-1}$, which is negligible and consistent with the result of comparison between lidar and radiosonde data."

R: Revised.

[revised manuscript text omitted]

---

## Author Response (AR2)

Dear Editor and Referees,

Thank you for your review of our manuscript. We greatly appreciate the substantial amount of time and effort that you dedicated to this review process.

We have revised the manuscript according to your comments and the point-by-point response is also attached in this file. The marked-up manuscript version showing the changes made is also provided as follow.

In order to facilitate access to the response, the corresponding page number ranges are listed below and you can also link to the corresponding section by clicking the hyperlink:

1. Response to reviewer 1: Page 1 - Page 3;
2. Response to reviewer 2: Page 3 - Page 4;
3. Marked-up manuscript version: Page 5 – Page 46;

Thanks again.

Kind regards.

Songhua Wu

----------------------------------------------Reviewer Comments----------------------------------------------

**Reviewer 1**

I'm not sure that the authors correctly obtain estimates for SNR (ratio of peak spectral amplitude to root-mean-square deviation of the noise spectrum) less than 5 dB. The fact is that at low SNR (for example, at SNR = 2 dB), the high probability that the spectral maximum is associated with noise, rather than with a signal. For the bandwidth of B = 100 MHz and the width of the time window T = 200 ns, the number of independent spectral channels L = B * T = 20. Accordingly, the number of peaks in the measured spectrum is not less than L / 2 = 10. When the amplitude of the signal peak in the spectrum is comparable with noise peaks, the probability that the signal peak is the maximum of the spectrum is about 1/10 = 0.1. Because of this, if the mean and the standard deviation of the noise spectrum are correctly obtained, the SNR estimate will be at least 2.5 (4 dB), even when there is no signal at all. The authors can verify this by using numerical simulation. The algorithm described by Eqs. (8) - (10) here does not help.

The authors have done a good job: an algorithm for processing of lidar measurements has been developed, which makes it possible to compensate for the measurement errors associated with the motion of the ship; joint measurements of the wind by a lidar and a radiosonde were carried out and a comparative analysis of the results of these measurements was made. The efficiency of the developed algorithm is shown. I think that the manuscript could be accepted for publication in AMT, if the authors will remove the results and analysis of the data received for the SNR less than 5 dB from the manuscript.

R: Regretfully, we don't very clearly understand "the number of spectral channel" you mentioned. Figure 3 in the revised manuscript shows the array of spectral $S(l\Delta f; k\Delta R)$. In the FFT processing, each range bin has L=100 spectral channels where the sampling frequency is 1 GHz and the point used for FFT calculation is 1024. In this case the frequency resolution $\Delta f \approx 0.98$ MHz and the corresponding velocity resolution is $\Delta V = 0.76$ ms$^{-1}$. The bandwidth $B_{100} = (L-1)\Delta f = 97.68$ MHz.

Figure 1 below shows the example of the profile of signal power $P_s(k\Delta R)$ and SNR $SNR(k\Delta R)$. The noise power $P_N$ is calculated using Eq. 8 and Eq. 10 in revised manuscript and $P_N = 0.24$ in this case. Taking the point at $H = 1021m$ for example shown in figure 1(b), the SNR at $H = 1021m$ is 5.1dB. Figure 2 shows the specific FFT spectral analysis at $H = 1021m$. It can be seen that the peak index $l_{peak}$ from the $\hat{S}(l\Delta f; k\Delta R)$ is $l_{peak} = 55$. The signal power $P_s(k\Delta R)$ can be calculated using Eq. 9 and $P_s(1021) = 0.78$ in this case. According to Eq.11, we obtain the $SNR(1021) = 10\log_{10}(\frac{0.78}{0.24}) \approx 5.1dB$. It can be seen that the signal peak can be obtained straightforward at $SNR = 5dB$.

[Figure]

Figure 1. An example of the profile of CDL (a) signal power $P_s(k\Delta R)$ and (b) SNR $SNR(k\Delta R)$ during shipborne experimental measurement.

[Figure]

Figure 2. The CDL measured array of the FFT spectra at height of 1021 m shown in figure 1.

**Reviewer 2**

The manuscript was significantly enhanced and revised. All my major comments and most of the minor comments were addressed in an extensive reply and also revised in the manuscript. I thank the authors for    this    very    good    reply.    I    have    only    a    few    technical    comments
1) Explain acronyms MEMS, RTK, CEP, CFD

R: Thanks for your suggestion, I didn't explain it clearly. The explanation has been added in the revised manuscript.

*MEMS: Micro-electromechanical Systems*

*RTK: Real Time Kinematic*

5    *CEP: Circular Error Probable*

*CFD: Computational Fluid Dynamics*

2) Mention distance hard target to lidar

R: Revised. The distance between hard target and lidar is about 300 m during the experiment.
3) "2-min" should be "2 minutes"

10   R:Revised.

4) Fig. 8: replace "RMSM" by "RMSE"

R: Revised.

[revised manuscript text omitted]

---

## Author Response (AR3)

Dear Editor,

Thank you for your review of our manuscript. We greatly appreciate the substantial amount of time and effort that you dedicated to this review process.

We have revised the manuscript according to your comments and the point-by-point response is also attached in this file. The marked-up manuscript version showing the changes made is also provided as follow.

In order to facilitate access to the response, the corresponding page number ranges are listed below and you can also link to the corresponding section by clicking the hyperlink:

1. Response to Editor: Page 1 - Page 3;

2. Marked-up manuscript version: Page 4 – Page 46;

Thanks again.

Kind regards.

Songhua Wu

-------------------------------------------Editor Comments-------------------------------------------------

The referee is assuming that, for a particular range bin, the FFT is calculated with a data time series of duration 200 ns, and that the sampling of that time series was done with a sampling interval of 5 ns. The assumed sampling interval is based on the bandwidth that you give, of 100 MHz. (As an aside, what is the significance of the subscript "100" on the B?) Presumably the reviewer has assumed that you then chose the sample rate to be 200 MS/s (twice the bandwidth according to the sampling theorem or Nyquist criterion). It is seems to me however that you are oversampling at 2 GS/s, based on your statement of the frequency resolution being approximately 1 MHz. It would be helpful if you were explicit about the sampling rate.

However the main part of the reviewer's objection centres around the fact that the uncertainty (variance) associated with the estimate of spectral power (or amplitude - you aren't very specific on this) in a single bin is equal to that estimate, because it must be assumed to be a random variable with Poisson statistics. Thus if you have the power in one bin sticking up above the average power in the bins, so that it is a factor

of two larger than the average (3dB), there is a significant probability that this is merely a background fluctuation, even if it is the highest bin. This assumes that you have just the one FFT for that particular range. Now if you average spectra the uncertainty (variance again) of the power in each bin (in the average) is reduced by a factor of the number of spectra used in the averaging. Then the reviewer's objection would not hold.Earlier in the paper you state that you do averaging in fact, but you don't really say how. This is important because if you average the signals before calculating the FFT, rather than averaging the spectra, the reviewer's objection holds.

There are other issues with the text that has been added. The word "spectral" is an adjective for one thing, which is somewhat distracting. The quantity S that it refers to is not well introduced, and it is not obvious or explained, what the reader is seeing in the new figure 3.

R: We didn't explain it clearly. The sampling rate in our study is $f = 1$ GHz, corresponding to the sampling interval of $T_s = 1$ ns. By appling the discrete Fourier transform to $M = 256$ samples, the spectrum estimate has a frequency resolution of $\Delta f = (MT_s)^{-1} = 3.906$ MHz and a velocity resolution of $\Delta V = (\lambda/2)\Delta f = 3.027$ ms$^{-1}$, where $\lambda = 1.55$ μm is the wavelength. In order to obtain the spectrum with better resolution, the interpolation method of adding 768 zeros to the 256 selected samples of the backscatter signal is used, thus the final frequency resolution is $\Delta f \approx 0.98$ MHz and the corresponding velocity resolution is $\Delta V = 0.76$ ms$^{-1}$.

Based on the heterodyne detection, the frequency of the local oscillator $f_0$ is optically mixed with the backscatter signal $f_0 + f_{IF} + \Delta f_D$, where $f_{IF}$ is the intermediate frequency and $\Delta f_D$ is the Doppler shift from atmospheric movement. Thus the beat signal of $f_{IF} + \Delta f_D$ is obtained in a detectable range $B$ of MHz. Figure 3 shows the array of the FFT spectrum $S(l\Delta f; k\Delta R)$ after zero padding interpolation obtained from the raw data measured with PCDL, where $l = 0, 1, 2, 3, ..., L-1$ is the spectral channel number and $L = 100$ and $k = 1, 2, 3..., K$ is the range bin number and $K = 104$. It is noted that the 100 spectral channel is symmetrically selected near the intermediate frequency with $B_{100} = (L-1)\Delta f = 97.68$ and corresponding radial velocity measurement range of $\pm 37.5$ ms$^{-1}$.

In our study the spectra were averaged firstly in order to reduce fluctuations from signal and noise components. The scattering particles in the sensing volume have a certain velocity distribution. When the SNR is high, it is easy to estimate the mean radial velocity from the FFT spectrum estimate. However, the evaluation of the maximal velocity with acceptable accuracy is not possible because of the strong fluctuations of the signal and noise components in the spectrum estimate. In order to reduce these fluctuations, the spectrum accumulation is used. In our study the pulse repetition rate is 10 KHz and the accumulation shots is 5000 for each radial velocity measurement.

Thanks for your suggestion, the "spectral" is replaced with "spctrum".

[revised manuscript text omitted]